# Prune-then-Quantize or Quantize-then-Prune? Understanding the Impact of Compression Order in Joint Model Compression

**Minjun Kim, Jaehyeon Choi, Hyunwoo Yang, Jongjin Kim, Jinho Song & U Kang**[*]
Seoul National University, Seoul, South Korea
{minjun.kim,ukang}@snu.ac.kr

## Abstract

What happens when multiple compression methods are combined—does the order in which they are applied matter? Joint model compression has emerged as a powerful strategy to achieve higher efficiency by combining multiple methods such as pruning and quantization. A central but underexplored factor in joint model compression is the compression order, or the sequence of different methods within the compression pipeline. Most prior studies have sidestepped the issue by assuming orthogonality between techniques, while a few have examined them only in highly constrained cases. Consequently, the broader role of compression order in shaping model performance remains poorly understood. In this paper, we address the overlooked problem of compression order and provide both theoretical and empirical analysis. We formulate the problem of optimizing the compression order and introduce the Progressive Intensity Hypothesis, which states that weaker perturbations should precede stronger ones. We provide theoretical guarantees showing that the relative benefit of one order increases with the underlying performance gap. Extensive experiments on both language and vision models validate the hypothesis, and further show its generality to broader setups such as multi-stage compression and mixed-precision quantization.

## 1 Introduction

*When combining pruning and quantization, which order leads to better model performance?* Although deep neural networks have achieved remarkable success across diverse domains, deploying them on edge devices remains challenging due to limited computational resources. To bridge this gap, network compression techniques (Deng et al., 2020; Liang et al., 2021; Zhu et al., 2024; Kim et al., 2025a) have been proposed, including pruning (Park et al., 2024; Song et al., 2024; Park et al., 2025b), quantization (Piao et al., 2022; Ashkboos et al., 2024b; Kim et al., 2025b), knowledge distillation (Kim et al., 2021b; Cho & Kang, 2022; Jeon et al., 2023), parameter sharing (Desai & Shrivastava, 2024; Wang et al., 2025a) and low-rank approximation (Jang et al., 2023; Li et al., 2025; Wang et al., 2025b). Recent studies highlight that combining these compression methods—known as *joint model compression*—achieves better trade-offs between compression ratio and model performance than applying them separately (Hawks et al., 2021; Wang et al., 2022; Shinde, 2024).

A critical yet underexplored issue in joint model compression is the *compression order*—the sequence in which individual compression methods are applied to the target model. As most of these techniques are not simultaneously applicable and should be executed sequentially (Wang et al., 2020; Kuzmin et al., 2023), identifying an optimal order can yield a "free lunch" by improving performance without any additional computation. Empirical findings (Huang et al., 2019; Hu et al., 2021; Qu et al., 2025) show that the performance of the compressed model is sensitive to the compression order, necessitating a deeper understanding of when and why certain orders work better.

However, the role of compression order has been largely overlooked by prior studies (Kurtic et al., 2022; Xiao et al., 2023; Liu et al., 2023). Most existing studies implicitly assume that compression order has no effect on the grounds of orthogonality, naïvely arguing that different techniques operate

---

[*]Corresponding Author.

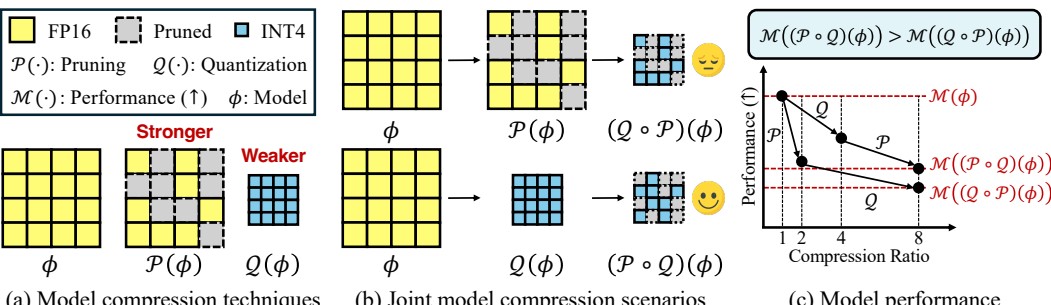

Figure 1: **The Progressive Intensity Hypothesis:** Given two compression techniques, we conjecture that compressed models perform better if the stronger method is applied after the weaker one. That said, the optimal order between pruning and quantization varies with their compression ratios.

independently (Kim et al., 2021a; Chitty-Venkata et al., 2023; Song et al., 2024; Motetti et al., 2024). Only a few works have examined the problem, and most of them merely offer empirical evidence confined to specific settings (Wang et al., 2020; Wu et al., 2023; Yu et al., 2023). A notable attempt (Harma et al., 2025) presents a theoretical framework, proving the non-orthogonality of pruning and quantization, concluding that pruning followed by quantization is always preferable. However, the scope of the work remains narrow and less practical, focusing only on magnitude-based pruning and max-scaled quantization (see Appendix D.5). To date, no study has systematically investigated the tendencies of compression order in general settings, neither empirically nor theoretically.

In this paper, we demonstrate that applying more aggressive compression algorithms at later stages yields superior performance. We first formulate the problem of *joint compression order optimization* (see Section 3.1 and Problem 1), and introduce *the Progressive Intensity Hypothesis*, which posits that ordering compression methods from weaker to stronger improves performance (see Hypothesis 1). Figure 1 offers a conceptual depiction of the proposed hypothesis. We validate our claim through both theoretical analysis and extensive experiments. Theoretically, we show that the advantage of the compression order grows monotonically with the performance gap between two methods under *disjoint sensitivity* (see Theorem 1 and Definition 5). In other cases, we define *interference* as an additional error from mutual interaction and investigate its influence (see Definition 6). Experimentally, we validate the hypothesis across both language and vision models, covering diverse model architectures, tasks, and compression scenarios (see Sections 5.2 and 5.3). Our analysis also considers how factors such as weight-update strategies and rotations affect the role of compression order (see Figures 4 and 5). Moreover, our results highlight that the hypothesis generalizes to broader paradigms, including multi-stage approaches and mixed-precision quantization (see Section 5.4).

Our contributions are summarized as follows:

- **Formulation.** We formally define the novel problem of optimizing the compression order in joint model compression (see Problem 1), and propose *the Progressive Intensity Hypothesis*, suggesting that stronger perturbations should be applied later to achieve better performance (see Hypothesis 1).
- **Theory.** We provide a theoretical analysis that quantifies the relationship between method interaction and order sensitivity. Specifically, we prove that the superiority of one ordering grows monotonically with the performance gap between the two methods (see Theorem 1).
- **Experiments.** Extensive and consistent experimental results across various domains, models, and tasks support our hypothesis (see Figures 3, 4, and 6). We further extend the problem to broader setups such as multi-stage compression and mixed-precision quantization (see Figures 7 and 10).

To the best of our knowledge, we are the first to both theoretically and experimentally analyze the impact of compression order in joint model compression under general and practical settings.

**Reproducibility.** All of our implementation and datasets are available at https://github.com/snudatalab/PQQP.

## 2 PRELIMINARIES AND RELATED WORKS

We briefly describe the preliminaries and related works on pruning, quantization, and joint model compression. The notations used throughout this paper are formally defined in Appendix A.

**Pruning and Quantization.** Compression[1] techniques aim to transform a pre-trained model $\phi$ as a more efficient version $\phi'$ while minimizing performance degradation (Xu & McAuley, 2023; Dantas et al., 2024; Liu et al., 2025a). This process inevitably introduces an error term $\delta(\cdot)$, representing the deviation between outputs of $\phi'$ and $\phi$, which typically increases with the compression ratio $C$. We define the compression ratio $C$ as the memory usage of $\phi$ divided by that of $\phi'$. Among various compression techniques $f(\phi; C)$, our work centers on two major forms: pruning and quantization.

Pruning $\mathcal{P}(\cdot)$ directly discards less important components of a model to achieve the desired compression ratio while retaining its most critical parts (Nova et al., 2023; Ashkboos et al., 2024a; Park et al., 2024). Based on the level of granularity, pruning methods fall into three categories: structured pruning (Song et al., 2024) removes entire structural elements such as layers, filters, or attention heads, semi-structured pruning (Xu et al., 2024) enforces fixed sparsity patterns (e.g., 2:4 sparsity) across tensors, and unstructured pruning (Frantar & Alistarh, 2023) prunes weights in a fully flexible manner. In the case of structured pruning at the layer level, the induced error $\delta_{\mathcal{P}}(\mathbf{W}_i, \mathbf{X}_i)$ is $-\mathbf{W}_i\mathbf{X}_i$ when pruning is applied to layer $l_i$ with weight $\mathbf{W}_i$ and activation $\mathbf{X}_i$, and $\mathbf{0}$ otherwise. The model achieves a compression ratio $C_{\mathcal{P}} = 1/(1-p)$ by pruning a fraction $p$ of weights.

Quantization $\mathcal{Q}(\cdot)$ reduces the bit precision used to represent weights and activations by encoding a high-bit network into a lower-bit format (Gholami et al., 2022). Common quantization techniques include uniform (Li et al., 2021), non-uniform (Zhao & Yuan, 2025), binary coding (Park et al., 2025a), and vector quantization (VQ) (Tseng et al., 2024). Although some techniques such as VQ focus only on weight quantization without compressing activations, our main scope is on compressing both for practical acceleration. A main challenge towards robust quantization is the activation outliers (Xiao et al., 2023; Lee et al., 2024), but recent rotation-based methods (Lin et al., 2024; Liu et al., 2025b) have largely overcome it. The layer-wise error by quantization $\mathcal{Q}(\cdot)$ for a layer $l_i$ with weight $\mathbf{W}_i$ and activation $\mathbf{X}_i$ is computed as $\delta_{\mathcal{Q}}(\mathbf{W}_i, \mathbf{X}_i) = \mathcal{Q}(\mathbf{W}_i)\mathcal{Q}(\mathbf{X}_i) - \mathbf{W}_i\mathbf{X}_i$, with a compression ratio $C_{\mathcal{Q}} = B_{orig}/B_{\mathcal{Q}}$ depending on the original $B_{orig}$ and target $B_{\mathcal{Q}}$ bit-widths.

**Joint Model Compression.** Joint compression combines two or more compression methods, achieving higher compression ratios while minimizing performance loss (Wang et al., 2020; Wu et al., 2023; Yu et al., 2023; Harma et al., 2025). These methods fall into two categories: co-designed and post-hoc frameworks. Although the former offers the benefit of integration-aware design, they tend to be method-specific and less adaptable to alternative configurations (Qu et al., 2025).

In contrast, combining independently designed techniques allows for method-agnostic pipelines that adapt easily to diverse architectures. Several pruning works (Kurtic et al., 2022; Xiao et al., 2023; Song et al., 2024) empirically confirm that such combinations with quantization are both feasible and beneficial. As independently designed techniques are applied one after another, the order of compression plays a key role. However, the impact of compression order has not been adequately examined in the current literature. We denote applying $f_1(\cdot)$ before $f_2(\cdot)$ as $f_1 \rightarrow f_2$ or $(f_2 \circ f_1)(\cdot)$.

## 3 JOINT COMPRESSION ORDER OPTIMIZATION

### 3.1 PROBLEM DEFINITION

We are given a pre-trained model and multiple compression techniques, each associated with a specific compression rate. The goal is to find the optimal order in which to sequentially apply these methods. An order is considered optimal if it minimizes the degradation in model performance. We quantify performance using a metric $\mathcal{M}(\cdot)$, where higher values indicate better outcomes (e.g., classification accuracy or the negative of perplexity). We provide the formal definition as Problem 1.

**Problem 1** (Joint Compression Order Optimization). *We have a pre-trained model $\phi$, a set of compression methods $\mathbb{F} = \{f_1(\cdot), f_2(\cdot), \cdots, f_n(\cdot)\}$, and a performance metric $\mathcal{M}(\cdot)$. For a set $\Pi = \{\pi : \mathbb{F} \rightarrow \mathbb{F} \mid \pi \text{ is bijective}\}$ of all permutations over $\mathbb{F}$, the goal is to find the optimal permutation $\pi^* \in \Pi$ that maximizes the performance of the compressed model: $\pi^* = \arg\max_{\pi \in \Pi} \mathcal{M}(\pi(\phi))$.*

---

[1]In the remainder of the paper, we use 'compression' to refer to 'model compression' for simplicity.

## 3.2 CHARACTERIZING COMPRESSION ATTRIBUTES

Two key attributes arise when characterizing compression in a general setting: granularity and intensity. Granularity refers to the smallest structural unit on which compression is applied, and intensity refers to how aggressively the method alters the model, measured by its impact on performance.

**Granularity of Compression.** Compression methods are not applied to the model as a whole, but rather operate locally on its individual components. We define compression granularity as the atomic level at which compression is performed. To formalize this notion, we begin by abstracting the model into a set of component types, such as layers, sublayers, or attention heads. We refer to these as *abstract types*, which define the structural units over which compression may act. For two abstract types $t_1$ and $t_2$, we say $t_1$ is *larger* than $t_2$ if $t_1$ strictly contains $t_2$ as a structural unit. Among all types that are larger than both $t_1$ and $t_2$, we define the *least upper type* $t_{\text{lut}}(t_1, t_2)$ as the smallest one. For a given model $\phi$, let $\mathcal{T}_\phi$ denote the set of abstract types; this set depends on the model architecture.

Each compression method $f(\cdot)$ may be applicable only to a subset of abstract types. We denote this subset by $\mathcal{T}_f \subseteq \mathcal{T}_\phi$, representing the structural levels at which $f(\cdot)$ can operate. For instance, layer-wise pruning in large language models is applicable only to units coarser than layers. Then, the granularity of $f(\cdot)$ is the smallest unit $t_f \in \mathcal{T}_f$ on which $f(\cdot)$ is applicable, as defined in Definition 1.

**Definition 1** (Compression Granularity). *For a model $\phi$ with a set $\mathcal{T}_\phi$ of abstract types and compression method $f(\cdot)$, the compression granularity $t_f := \arg\min_{t \in \mathcal{T}_f} |t|$, where $\mathcal{T}_f \subseteq \mathcal{T}_\phi$ denotes the set of abstract types on which $f(\cdot)$ operates, and $|t|$ denotes the structural size of type $t$.*

**Intensity of Compression.** Compression methods affect the model differently even at identical compression ratios, so comparing their intensities directly is challenging. To assess compression strength, we introduce three concepts grounded in performance degradation: performance gap $\mathcal{G}(f_1, f_2)$, compression equivalent ratio $C_f^*$, and compression order advantage $\mathcal{A}(f_1 \to f_2)$.

Performance differences between two methods $f_1(\cdot; C_1)$ and $f_2(\cdot; C_2)$, each applied at its respective compression ratios $C_1$ and $C_2$, provide a direct measure of their relative intensity. We call this the *performance gap* $\mathcal{G}(\phi, \mathcal{M}; f_1(\cdot; C_1), f_2(\cdot; C_2))$, or simply $\mathcal{G}(f_1, f_2)$, as defined in Definition 2. If $\mathcal{G}(f_1, f_2) > 0$, we refer to $f_2(\cdot; C_2)$ as the stronger compression and $f_1(\cdot; C_1)$ as the weaker one.

**Definition 2** (Performance Gap). *Given a model $\phi$, a performance metric $\mathcal{M}(\cdot)$, and two compression methods $f_1(\cdot; C_1)$ and $f_2(\cdot; C_2)$, the performance gap between two methods $\mathcal{G}(\phi, \mathcal{M}; f_1(\cdot; C_1), f_2(\cdot; C_2)) := \mathcal{M}(f_1(\phi; C_1)) - \mathcal{M}(f_2(\phi; C_2))$.*

Although $\mathcal{G}(\cdot)$ offers a clear pairwise comparison, its values in metric units are difficult to interpret and may grow rapidly as the compression ratio increases. Alternatively, mapping methods onto a common scale allows for direct comparison at the level of compression ratios. While multiple choices exist for the baseline method, we select quantization as it exhibits the best performance across diverse models, thereby offering the widest range. Accordingly we define the Compression Equivalent Ratio (CER) $C^*(f_1(\cdot), \mathcal{Q}, C)$, or simply $C_{f_1}^*$, which expresses the effect of method $f_1(\cdot; C)$ at ratio $C$ as an equivalent ratio of quantization $\mathcal{Q}(\cdot)$, as Definition 3. In other words, starting from a 16-bit model, a compression method $f(\phi; C)$ with $C_f^* = 2$ achieves the same performance as 8-bit quantization. Note that CER of quantization $\mathcal{Q}(\cdot)$ is naturally equal to its own compression ratio (i.e., $C_\mathcal{Q}^* = C_\mathcal{Q}$). We adopt a straightforward approach by computing CER through linear interpolation. For instance, $f(\cdot)$ achieving $\mathcal{M}(f; C) = 65\%$ accuracy maps to $C_f^* = 3$ when quantization $\mathcal{Q}(\cdot)$ yields $\mathcal{M}(\mathcal{Q}; C_\mathcal{Q} = 2) = 70\%$ and $\mathcal{M}(\mathcal{Q}; C_\mathcal{Q} = 4) = 60\%$ accuracy, respectively.

**Definition 3** (Compression Equivalent Ratio). *Given a model $\phi$, a performance metric $\mathcal{M}(\cdot)$, a compression method $f(\cdot)$, a quantization method $\mathcal{Q}(\cdot)$, and a compression ratio $C$, the compression equivalent ratio $C^*(f(\cdot), \mathcal{Q}, C) := C'$ such that $\mathcal{M}(\mathcal{Q}(\phi; C')) = \mathcal{M}(f(\phi; C))$.*

Until now our discussion is limited to single methods; but when multiple methods are applied, how should intensity be defined? Our scope centers on measuring how intensity changes by compression order. Accordingly, we capture the gain from applying $f_1(\cdot)$ before $f_2(\cdot)$ over the reverse as compression order advantage $\mathcal{A}(\phi, \mathcal{M}; f_1(\cdot) \to f_2(\cdot))$, or simply $\mathcal{A}(f_1 \to f_2)$, as Definition 4.

**Definition 4** (Compression Order Advantage). *Given a model $\phi$, a performance metric $\mathcal{M}(\cdot)$, and two compression methods $f_1(\cdot; C_1)$ and $f_2(\cdot; C_2)$, the compression order advantage $\mathcal{A}(\phi, \mathcal{M}; f_1(\cdot; C_1) \to f_2(\cdot; C_2)) := \mathcal{G}(f_1 \to f_2, f_2 \to f_1) = \mathcal{M}((f_2 \circ f_1)(\phi)) - \mathcal{M}((f_1 \circ f_2)(\phi)).$*

### 3.3 THE PROGRESSIVE INTENSITY HYPOTHESIS

Our goal is to uncover general patterns in how compression order affects the model performance in joint compression scenarios. While prior works have focused primarily on isolated settings, we seek to establish a broadly applicable principle. To this end, we propose *the Progressive Intensity Hypothesis*, which posits that applying stronger compression methods at later stages generally yields better performance. We formalize this hypothesis for a pair of methods in Hypothesis 1, which serves as the main focus of our analysis; its extension to multiple methods is presented in Appendix B.3.

**Hypothesis 1** (The Progressive Intensity Hypothesis). *Let $f_1(\cdot; C_1)$ and $f_2(\cdot; C_2)$ be two compression methods applied to a model $\phi$. Then, the compression order advantage $\mathcal{A}(f_1 \rightarrow f_2)$ grows monotonically with the performance gap $\mathcal{G}(f_1, f_2)$, or equivalently with the CER difference $C^*_{f_2} - C^*_{f_1}$.*

As an example, if methods $f_1(\cdot)$ and $f_2(\cdot)$ yields $\mathcal{M}(f_1; C_1) = 75\%$ and $\mathcal{M}(f_2; C_2) = 70\%$ accuracy, respectively (i.e., $\mathcal{G}(f_1, f_2) = 5\%$p), the compression order advantage $\mathcal{A}(f_1 \rightarrow f_2)$ is mild; replacing $C_2$ into $C'_2$ at $\mathcal{M}(f_2; C'_2) = 60\%$ accuracy (i.e., $\mathcal{G}(f_1, f_2) = 15\%$p) results in a larger advantage.

## 4 THEORETICAL ANALYSIS

We theoretically analyze how compression order affects the model performance. We introduce disjoint selectivity to isolate order-dependent units, and prove in Theorem 1 that only these units determine the performance gap. We then show in Theorem 2 that Hypothesis 1 holds due to the reduction of order-dependent units. We later extend to non-disjoint cases in which interference occurs. Consistent with earlier works (Harma et al., 2025), we investigate each unit, relying on Assumption 1.

**Assumption 1.** *Given a model $\phi$ with a set $\mathbb{L}$ of layers, performance metric $\mathcal{M}(\cdot)$, a compression method $f(\cdot)$, and the layer-wise reconstruction loss $\delta_f(l_i)$, assume the followings:*

- ***Layer-wise independence.** The reconstruction error at one layer does not affect the reconstruction error at another: $\forall l_i, l_j \in \mathbb{L}, \ i \neq j: \ \partial\,\delta_f(l_i)/\partial\,\delta_f(l_j) = 0$.*
- ***Error-performance trade-off.** Model performance is inversely related to total reconstruction error: $\exists \beta > 0, \ \mathcal{M}(\phi) - \mathcal{M}(f(\phi)) = \beta \cdot \sum_{l_i \in \mathbb{L}} \|\delta_f(l_i)\|_F^2$.*

**Disjoint Selectivity.** Sequential application of two compression methods leads to two distinct scenarios: either there exist units altered by both methods, or all units are exclusively assigned to one. We define the latter scenario as the case where *disjoint selectivity* holds, as in Definition 5. This means that while the assignment may vary with order, each unit is ultimately handled by only one method.

**Definition 5** (Disjoint Selectivity). *Given a model $\phi$, two compression methods $f_1(\cdot)$ and $f_2(\cdot)$ with respective granularities $t_{f_1}$ and $t_{f_2}$, disjoint selectivity holds if $\forall u_i \in \mathbb{U}(\phi; t_{lut}(t_{f_1}, t_{f_2})), \ \forall \pi \in \{f_1 \circ f_2, f_2 \circ f_1\}, \ \mathbb{D}_{u_i}^{f_1}(\pi) + \mathbb{D}_{u_i}^{f_2}(\pi) = 1$, where $\mathbb{U}(\phi; t)$ is the set of all units of model $\phi$ at granularity $t$, and $\mathbb{D}_u^f(\pi)$ denotes whether $f(\cdot)$ modifies unit $u$ under the order $\pi$ (i.e., 1 if modified, 0 otherwise).*

Under disjoint selectivity, the compression order advantage $\mathcal{A}(f_1 \rightarrow f_2)$ is proportional to the cumulative sum of error difference $g(\cdot)$ across units assigned differently depending on the order as formulated in Theorem 1. The underlying intuition is that the performance gap rises solely from units whose assignment varies with the order; for others, the error remains invariant and thus cancels out. To illustrate, consider units $u_1$, $u_2$, and $u_3$ and compression methods $f_1(\cdot)$ and $f_2(\cdot)$. If $u_1$ is always handled by $f_1(\cdot)$ regardless of the order, while $u_2$ and $u_3$ are assigned differently depending on the order, then the advantage $\mathcal{A}(f_1 \rightarrow f_2)$ is proportional to error difference of units $u_2$ and $u_3$.

**Theorem 1** (Compression Order Advantage under Disjoint Selectivity). *Suppose we compress a model $\phi$ with two compression methods $f_1(\cdot)$ and $f_2(\cdot)$ with respective granularities $t_{f_1}$ and $t_{f_2}$, where disjoint selectivity holds. Then, under Assumption 1, the compression order advantage*

$$\mathcal{A}(f_1 \rightarrow f_2) = \mathcal{M}((f_2 \circ f_1)(\phi)) - \mathcal{M}((f_1 \circ f_2)(\phi)) \text{ equals to } \beta \cdot \left( \sum_{u_i \in \mathbb{G}_2} g(u_i) - \sum_{u_i \in \mathbb{G}_1} g(u_i) \right),$$

*where $\beta$ is the coefficient between the model performance and total error induced from Assumption 1, $g(u_i) = \left\|\delta_{f_1}(u_i)\right\|_F^2 - \left\|\delta_{f_2}(u_i)\right\|_F^2$ is the error gap according to the method applied, and $\mathbb{G}_1 = \{u \mid \mathbb{D}_u^{f_1}(f_2 \circ f_1) = 1, \ \mathbb{D}_u^{f_1}(f_1 \circ f_2) = 0\}$ and $\mathbb{G}_2 = \{u \mid \mathbb{D}_u^{f_1}(f_2 \circ f_1) = 0, \ \mathbb{D}_u^{f_1}(f_1 \circ f_2) = 1\}$ are groups of order-dependent units.*

*Proof.* Refer to Appendix B.1. □

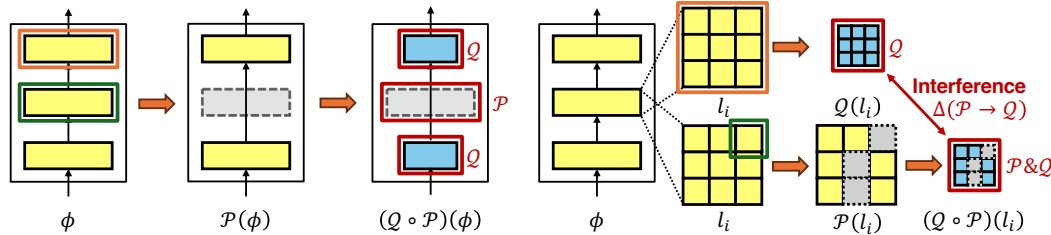

(a) Pruning granularity ≥ Quantization granularity  (b) Pruning granularity < Quantization granularity

Figure 2: A case study of pruning $\mathcal{P}(\cdot)$ and quantization $\mathcal{Q}(\cdot)$ on model $\phi$. (a) if pruning granularity (green) is coarser or equal to quantization granularity (orange), disjoint selectivity holds. (b) Otherwise, partial removal of quantization units by pruning introduces extra error, termed *interference* $\Delta$.

**Monotonicity.** Under disjoint selectivity, we show that Hypothesis 1 holds when the two compression methods are *well-designed*—that is, minimally disruptive to the model. We examine this through a case study on pruning and quantization. We assume a favorable scenario where pruning is configured to induce minimal degradation, and quantization introduces symmetric, zero-mean errors centered at the original values. These assumptions are formalized in Assumption 2.

**Assumption 2.** *Given a model $\phi$ with a set $\mathbb{L}$ of layers and performance metric $\mathcal{M}(\cdot)$, assume that:*

- **Well-designed pruning** $\mathcal{P}(\cdot)$**.** *The pruning method is chosen from the set of pruning strategies that aim to preserve the model performance: $\mathcal{P}(\cdot) \in \mathbb{P}(C_{\mathcal{P}})$ where $\mathbb{P}(C_{\mathcal{P}}) = \{\mathcal{P}_i(\cdot) | C(\mathcal{P}_i(\phi)) = C_{\mathcal{P}}, \ \mathcal{M}(\phi) - \mathcal{M}(\mathcal{P}_i(\phi)) \leq \delta\}$ denotes the set of pruning strategies that satisfy the target ratio $C_{\mathcal{P}}$ while keeping performance degradation within a small budget $\delta$.*

- **Well-designed quantization** $\mathcal{Q}(\cdot)$**.** *For all layers, quantized outputs follow a symmetric distribution around the original values: $\forall l_i \in \mathbb{L}, \mathcal{Q}(\mathbf{W}_i)\mathcal{Q}(\mathbf{X}_i) \sim \mathcal{N}(\mathbf{W}_i\mathbf{X}_i, \sigma_{\mathcal{Q}}^2\mathbf{I})$, where $\mathcal{N}(\cdot)$ is the Gaussian distribution. The quantization error is negligible (i.e., $\mathcal{Q}(\mathbf{W}_i)\mathcal{Q}(\mathbf{X}_i) - \mathbf{W}_i\mathbf{X}_i \ll \mathbf{W}_i\mathbf{X}_i$).*

Theorem 2 states that when disjoint selectivity holds and the compression methods are well-designed, $\mathcal{A}(\mathcal{Q} \to \mathcal{P})$ increases monotonically with CER difference[2] $C_{\mathcal{P}}^* - C_{\mathcal{Q}}$ for fixed $C_{\mathcal{P}}$. Note that as $C_{\mathcal{Q}}$ decreases (i.e., $\mathcal{M}(\mathcal{Q}(\phi))$ increases), both CER difference $C_{\mathcal{P}}^* - C_{\mathcal{Q}}$ and performance gap $\mathcal{G}(\mathcal{Q}, \mathcal{P})$ increase monotonically. We show that $\mathcal{A}(\mathcal{Q} \to \mathcal{P})$ increases in this setting because the gap depends solely on order-dependent units under Theorem 1. We discuss the impact of $C_{\mathcal{P}}$ in Appendix D.6.

**Theorem 2** (Monotonicity). *Suppose we compress a model $\phi$ with pruning $\mathcal{P}(\cdot)$ and quantization $\mathcal{Q}(\cdot)$, where disjoint selectivity holds. Then, under Assumptions 1 and 2, given performance metric $\mathcal{M}(\cdot)$ and two pairs of compression ratios $(C_{\mathcal{P}_1}, C_{\mathcal{Q}_1})$ and $(C_{\mathcal{P}_1}, C_{\mathcal{Q}_2})$, if CER difference increases*
$$C_{\mathcal{P}_1}^* - C_{\mathcal{Q}_1} > C_{\mathcal{P}_1}^* - C_{\mathcal{Q}_2},$$
*then, the compression order advantage increases monotonically:*
$$\mathcal{A}(\phi, \mathcal{M}; \mathcal{Q}(\cdot; C_{\mathcal{Q}_1}) \to \mathcal{P}(\cdot; C_{\mathcal{P}_1})) \geq \mathcal{A}(\phi, \mathcal{M}; \mathcal{Q}(\cdot; C_{\mathcal{Q}_2}) \to \mathcal{P}(\cdot; C_{\mathcal{P}_1})).$$

*Proof.* Refer to Appendix B.2. □

**Granularity and Interference.** Disjoint selectivity does not always hold in practical joint compression settings for pruning and quantization. As pruning operates by fully discarding or keeping each unit, it always satisfies disjoint selectivity. In contrast, quantization satisfies this condition only when its granularity is finer than or equal to that of pruning. Figure 2 illustrates this: (a) if $t_{\mathcal{P}} \geq t_{\mathcal{Q}}$, disjoint selectivity is preserved as pruning removes entire quantization units. However, (b) if $t_{\mathcal{P}} < t_{\mathcal{Q}}$, pruning may partially eliminate a quantization unit, introducing regions where both methods interfere.

In general joint compression of two methods $f_1(\cdot)$ and $f_2(\cdot)$, this violation of disjoint selectivity introduces additional error, which we define as *interference* $\Delta(\phi; f_1 \to f_2)$, or simply $\Delta(f_1 \to f_2)$ in Definition 6. Intuitively, interference quantifies how one method disturbs the behavior of the other.

**Definition 6** (Interference). *Given a model $\phi$ and two methods $f_1(\cdot)$ and $f_2(\cdot)$, the interference*
$$\Delta(\phi; f_1 \to f_2) := \sum_{u \in \mathbb{X}} \big(\delta_{f_2 \circ f_1}(u) - \delta_{f_2}(u)\big), \ where \ \mathbb{X} = \mathbb{U}(\phi; t_{f_2}) \ \cap \ \{u \mid \mathbb{D}_u^{f_2}(f_2 \circ f_1) = 1\},$$
*set $\mathbb{U}(\phi; t)$ contains all units of model $\phi$ at type $t$, $\mathbb{D}_u^f(\pi)$ indicates whether unit $u$ is modified by $f(\cdot)$ under order $\pi$ (1 if modified, 0 otherwise), and $\delta_{f(\cdot)}(u)$ denotes the error on $u$ after applying $f(\cdot)$.*

---

[2]By definition, quantization serves as the baseline scale, so its CER equals its own compression ratio (i.e., $C_{\mathcal{Q}}^* = C_{\mathcal{Q}}$). Therefore, $C_{\mathcal{P}}^* - C_{\mathcal{Q}}$ represents the CER difference between pruning and quantization.

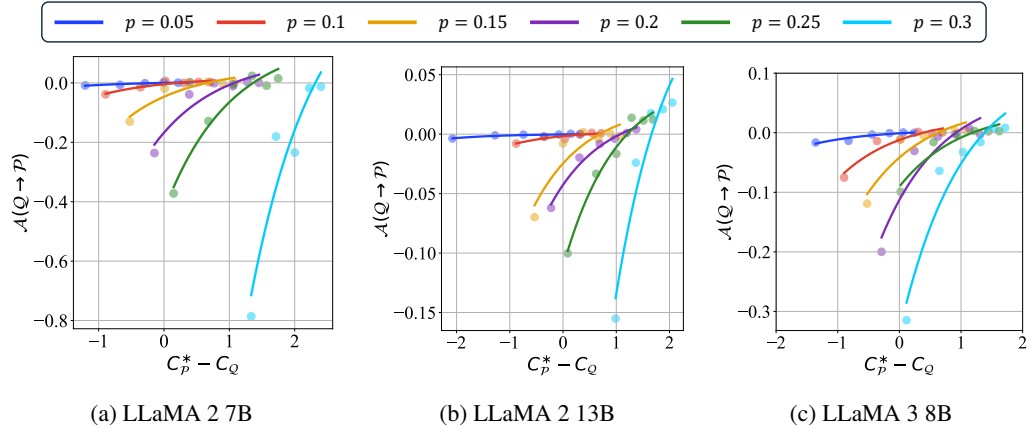

(a) LLaMA 2 7B          (b) LLaMA 2 13B          (c) LLaMA 3 8B

Figure 3: Across diverse language models, the compression order advantage $\mathcal{A}(\mathcal{Q} \to \mathcal{P})$ increases monotonically with the CER difference $C_{\mathcal{P}}^* - C_{\mathcal{Q}}$. See Section 5.2 for details.

Interference may or may not occur, depending on its applied techniques. A notable example is mixed-precision quantization, where treating each bit-width quantization as a distinct method satisfies disjoint selectivity, thereby avoiding interference. In our primary focus of pruning and quantization, the magnitude of interference is determined by the pruning ratio $p$. When interference occurs, quantization operates on weights altered by pruning, thus the deviation from the original distribution scales with the pruning ratio. As pruning ratio $p$ increases, a larger portion of weights is removed before quantization, leading to stronger interference. This additional error depends only on pruning and enters additively into the order advantage, and thus remains independent of quantization strength while preserving the monotonic trend. Consequently, while exact outcomes may vary, $\mathcal{A}(f_1 \to f_2)$ remains a monotonic function of $C_{\mathcal{P}}^* - C_{\mathcal{Q}}$ even under interference. In conclusion, Hypothesis 1 holds under both disjoint and interfering scenarios, highlighting its general validity.

## 5 EXPERIMENTAL FINDINGS

We empirically validate our hypothesis in joint compression scenarios, starting with pruning and quantization on language and vision models. We then extend to general pipelines beyond them.

### 5.1 EXPERIMENTAL SETUP

We briefly introduce the experimental setup. Further setups are detailed in Appendix C.

**Setup.** For language models, we focus on decoder-only LLMs, mainly LLaMA (Touvron et al., 2023) models. The main metric is the negative of perplexity on WikiText-2 (Merity et al., 2017) dataset; results on commonsense reasoning tasks appear in Appendix D.3. For vision models, we evaluate the classification accuracy of ResNet-18 (He et al., 2016) (CNNs) and DeiT-Base (Touvron et al., 2021) (ViTs) models on ImageNet (Deng et al., 2009) dataset.

**Baselines.** We evaluate three pruning (SparseGPT (Frantar & Alistarh, 2023), Wanda (Sun et al., 2024), and SLEB (Song et al., 2024)) and four quantization methods (RTN (Gupta et al., 2015), OPTQ (Frantar et al., 2023), QuaRot (Ashkboos et al., 2024b), and QuaRot + OPTQ) for language models. For vision models, we apply PRACTISE (Wang & Wu, 2023) and N2UQ (Liu et al., 2022) for CNNs, and adopt SAViT (Chuanyang et al., 2022) and RepQ-ViT (Li et al., 2023) for ViTs.

### 5.2 ANALYSIS ON LANGUAGE MODELS

We verify whether Hypothesis 1 holds for language models. Then, we analyze the effect of weight updates and rotations in quantization, and investigate the impact of pruning granularity on interference.

**Compression Order Advantage by CER Differences.** We analyze how the compression order advantage $\mathcal{A}(\mathcal{Q} \to \mathcal{P})$ varies with CERs for SparseGPT ($\mathcal{P}(\cdot)$) and QuaRot ($\mathcal{Q}(\cdot)$) across three models: LLaMA 2 7B, 13B, and LLaMA 3 8B. Figure 3 confirms that $\mathcal{A}(\mathcal{Q} \to \mathcal{P})$ increases monotonically with the CER difference for all three models. Each point in the figure corresponds to a compression

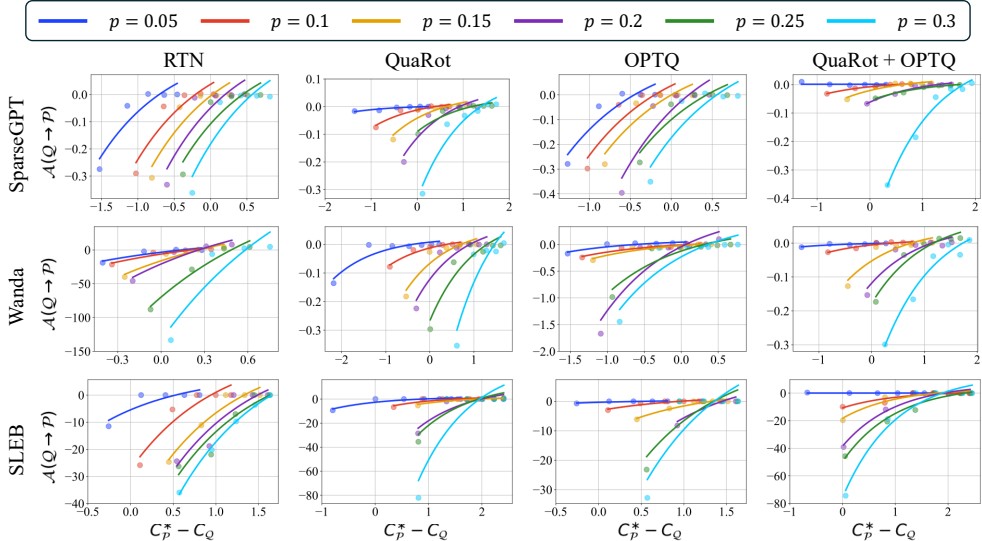

Figure 4: Compression order advantage $\mathcal{A}(\mathcal{Q} \to \mathcal{P})$ against CER difference $C_{\mathcal{P}}^* - C_{\mathcal{Q}}$ for three pruning $\mathcal{P}(\cdot)$ and four quantization $\mathcal{Q}(\cdot)$ methods on a LLaMA 3 8B model. Our hypothesis consistently holds for language models regardless of pruning granularity, rotation, and weight updates.

ratio pair $(C_{\mathcal{P}}, C_{\mathcal{Q}})$, defined by pruning ratio $p \in [0.05, 0.1, 0.15, 0.2, 0.25, 0.3]$ and quantization bit-width $B_{\mathcal{Q}} \in [4, 5, 6, 7, 8]$. We fit an exponential curve per pruning ratio $p$, reflecting the underlying trend. In this setting, the x-axis $C_{\mathcal{P}}^* - C_{\mathcal{Q}}$ captures the difference between the intrinsic intensities of pruning and quantization, with both $C_{\mathcal{P}}^*$ and $C_{\mathcal{Q}}$ calibrated on the original model $\phi$. Consistent trends across diverse language model architectures and scales supports the validity of Hypothesis 1; see Appendices D.2 and D.1 for results on encoder-based and other decoder-only models, respectively.

> **Finding 1.** The Progressive Intensity Hypothesis holds across diverse language models of varying scales and architectures, showing that stronger compression should be applied later.

**Weight Updates and Rotation-based Transformations.** We investigate the hypothesis under practical techniques such as weight-updates and rotations. Figure 4 shows that Hypothesis 1 consistently holds across diverse combinations of methods. Our framework is agnostic to the type of compression methods; weight updates and rotations reduce quantization error, thereby increasing $C_{\mathcal{P}}^*$.

> **Finding 2.** The hypothesis generalizes beyond specific design choices of pruning and quantization, remaining robust under weight-update and rotation methods.

An intriguing phenomenon arises in pruning rotation-based methods: applying pruning after rotation results in a drastic performance drop compared to pruning without rotation. Figure 5 illustrates the perplexity changes of LLaMA 3 8B model pruned by SparseGPT, depending on the Hadamard rotation from QuaRot. Increasing the pruning ratio amplifies the discrepancy between rotated and non-rotated settings, as pruning is applied without accounting for rotation. This effect emerges because rotation may introduce two types of errors: *matrix-wise errors* from residual components and *element-wise errors* from altered pruning decisions. We further discuss the details in Appendix D.4 and Table 4. As rotation may intensify pruning, it is essential to design pruning approaches compatible with rotation-based quantization, the emerging *de facto* standard.

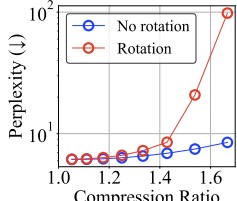

Figure 5: Rotation impact on pruning. See Section 5.2 for details.

> **Finding 3.** Rotation amplifies pruning effects, underscoring the necessity of designing rotation-aware pruning methods.

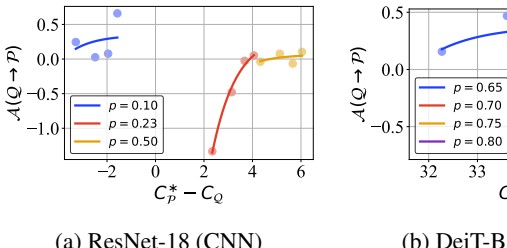

(a) ResNet-18 (CNN)    (b) DeiT-Base (ViTs)

Figure 6: The Progressive Intensity Hypothesis holds across diverse vision models. See Section 5.3 for details.

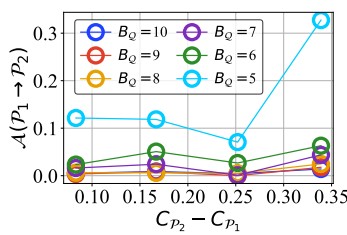

Figure 7: Multi-stage compression results on LLaMA 3 8B model.

**Pruning Granularity and Interference.** We verify the presence of interference by comparing results across different pruning granularities. Table 1 reports $\mathcal{A}(\mathcal{Q} \rightarrow \mathcal{P})$ across different quantization bit-width $B_{\mathcal{Q}}$ when applying two 5% pruning methods with QuaRot. For SLEB, which applies structured pruning at sublayer level, there exists a regime where no layers differ in their pruning status across orders, leading to an exact $\mathcal{A}(\mathcal{Q} \rightarrow \mathcal{P})$ of zero (i.e., no interference). By contrast, SparseGPT, as an unstructured pruning method, exhibits interference in low $C_{\mathcal{Q}}$ ranges. Notably, empirical results suggest that interference also exhibits a monotonic trend regarding $C_{\mathcal{Q}}$.

Table 1: $\mathcal{A}(\mathcal{Q} \rightarrow \mathcal{P})$ by quantization ratio $C_{\mathcal{Q}}$.

| $C_{\mathcal{Q}}$ $(B_{\mathcal{Q}})$ | SparseGPT | SLEB |
|---|---|---|
| 1.78 (9) | 0.002 | 0 |
| 2.00 (8) | 0.001 | 0 |
| 2.28 (7) | -0.003 | 0 |
| 2.68 (6) | -0.013 | 0 |
| 3.20 (5) | -0.017 | -0.057 |
| 4.00 (4) | -49.899 | -9.379 |

> **Finding 4.** Pruning granularity determines interference: structured pruning shows no interference in early regimes, while unstructured pruning exhibits monotonic interference.

## 5.3 ANALYSIS ON VISION MODELS

**CNN and ViT Models.** We verify whether Hypothesis 1 holds for vision models, focusing on CNNs and ViTs. In Figure 6, we analyze the behavior of ResNet-18 and DeiT-Base models under PRACTISE ($\mathcal{P}(\cdot)$) and N2UQ ($\mathcal{Q}(\cdot)$), and SAViT ($\mathcal{P}(\cdot)$) and RepQ-ViT ($\mathcal{Q}(\cdot)$) methods, respectively. The results confirm that both $\mathcal{A}(\mathcal{Q} \rightarrow \mathcal{P})$ and $C_{\mathcal{P}}^{*} - C_{\mathcal{Q}}$ increase monotonically for both models, regardless of the pruning or quantization configurations. Notably, the compression order advantage is substantially larger than that observed in language models, where it is often marginally positive.

> **Finding 5.** Vision models consistently satisfy the Progressive Intensity Hypothesis regardless of architecture and applied compression techniques.

## 5.4 BEYOND PRUNING AND QUANTIZATION: TOWARD GENERAL PIPELINES

We extend the Progressive Intensity Hypothesis to general compression pipelines. These results align with the $n$-method ordering formulation in Appendix B.3.

**Multi-stage Compression.** Pruning is generally performed in multiple stages to mitigate performance degradation. In Figure 7, we investigate the impact of compression order by alternately applying SparseGPT ($\mathcal{P}(\cdot)$) and QuaRot ($\mathcal{Q}(\cdot)$) to the LLaMA 3 8B model where the sum of pruning ratios $p_1 + p_2 = 0.3$ (e.g., $\mathcal{P}(\cdot; C_{\mathcal{P}_1}) \rightarrow \mathcal{Q}(\cdot) \rightarrow \mathcal{P}(\cdot; C_{\mathcal{P}_2})$). Our results consistently demonstrate positive advantages, indicating that stronger pruning placed later improves performance under fixed quantization, confirming that our hypothesis holds not only for two stages but also for multiple ones.

> **Finding 6.** Beyond pairwise orders, the hypothesis holds in practical multi-stage compression, indicating that scheduling stronger compression later in the sequence yields higher accuracy.

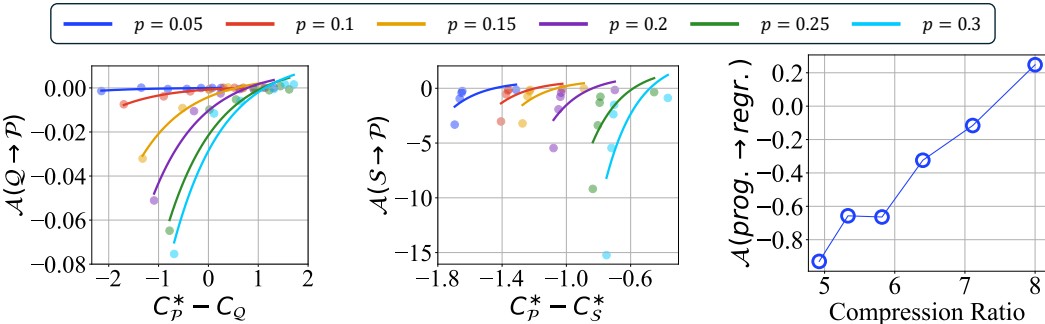

Figure 8: Joint compression results with LoRA adapters.

Figure 9: Performance under pruning and parameter sharing.

Figure 10: The impact of compression order in MPQ.

**Parameter Efficient Fine-tuning.** Parameter Efficient Fine-Tuning (PEFT), which introduces lightweight low-rank adapters to mitigate compression-induced performance degradation, has recently become a widely adopted practical approach. We investigate whether the proposed principle remains valid in scenarios where PEFT is applied alongside pruning and quantization. Figure 8 confirms that the same pattern holds for LLaMA 3 8B when combined with SparseGPT ($\mathcal{P}(\cdot)$), RTN ($\mathcal{Q}(\cdot)$), and LoRA (Hu et al., 2022) (PEFT). Applying LoRA after quantization produced a similar corrective effect to rotation, effectively compensating quantization-induced performance loss. Overall, the progressive intensity hypothesis remains robust under practical training pipelines that include PEFT.

> **Finding 7.** PEFT preserves the hypothesis that stronger compression should be applied later, as post-quantization LoRA effectively restores accuracy and maintains the expected ordering.

**Parameter Sharing.** Beyond pruning and quantization, parameter sharing $\mathcal{S}(\cdot)$ is an independent compression technique that ties multiple layers into a unified set of weight parameters. We conduct joint compression experiments with pruning and parameter sharing to assess whether the principle generalizes beyond the pruning–quantization setting. In Figure 9, results on LLaMA 2 7B model with Basis Sharing (Wang et al., 2025a) ($\mathcal{S}(\cdot)$) and magnitude-based pruning (Han et al., 2015) ($\mathcal{P}(\cdot)$) confirm that the same ordering effect emerges as well.

> **Finding 8.** Joint compression with parameter sharing also follows the hypothesis: placing the stronger operation later yields better performance.

**Mixed-precision Quantization.** As discussed in Section 4, Mixed Precision Quantization (MPQ) can be formulated as a joint compression problem where each bit-width allocation acts as a separate compression method, satisfying disjoint selectivity. Figure 10 illustrates the effect of compression order in MPQ, where we sequentially allocate quantization bit-widths following HAWQ-V2 (Dong et al., 2020) on ResNet-18 model under a fixed average bit-width (i.e., identical overall compression ratio), comparing progressive (prog.; 8→2 bits) and regressive (regr.; 2→8 bits) sequential allocations. As the total compression ratio increases, progressive allocation increasingly outperforms regressive allocation in terms of $\mathcal{A}(\text{prog.} \to \text{regr.})$, supporting our hypothesis under MPQ settings.

> **Finding 9.** The Progressive Intensity Hypothesis also holds in MPQ, with progressive bit allocation outperforming regressive allocation since lower-bit quantization acts as stronger compression.

## 6 CONCLUSION

We address the under-explored problem of joint compression order optimization and provide both theoretical and experimental evidences. The Progressive Intensity Hypothesis (Hypothesis 1) offers a simple yet powerful rule: weaker perturbations first, stronger ones later. Future works include investigating interference in more complex pipelines, providing explicit predictive rules, and automating compression order selection (see Appendix D.7).

## ACKNOWLEDGMENTS

This work was supported by Institute of Information & communications Technology Planning & Evaluation (IITP) grant funded by the Korea government (MSIT) [No.RS-2020-II200894, Flexible and Efficient Model Compression Method for Various Applications and Environments], [No.RS-2021-II211343, Artificial Intelligence Graduate School Program (Seoul National University)], [No.RS-2024-00509257, Global AI Frontier Lab], and [No.RS-2025-25442338, AI star Fellowship Support Program(Seoul National University)]. This work was supported by Youlchon Foundation. The Institute of Engineering Research at Seoul National University provided research facilities for this work. The ICT at Seoul National University provided research facilities for this study. U Kang is the corresponding author.

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

## A  Notation

We summarize the frequently used notations in the paper as Table 2.

Table 2: Frequently used notations.

| Symbol | Description |
|---|---|
| $\phi$ | A pre-trained model |
| $\phi'$ | A compressed model |
| $f(\cdot) \in \mathbb{F}$ | Compression method from a set $\mathbb{F}$ |
| $C$ | Compression ratio |
| $\mathcal{M}(\cdot)$ | Performance metric |
| $\mathbf{W}_i, \mathbf{X}_i$ | Weight and activation matrices of layer $l_i$, respectively |
| $\pi \in \Pi$ | Compression order from a set $\Pi$ of all possible permutations |
| $\mathcal{P}(\cdot), \mathcal{Q}(\cdot)$ | Pruning and quantization methods, respectively |
| $p$ | Pruning ratio $(C_{\mathcal{P}} = 1/(1-p))$ |
| $B_{orig}, B_{\mathcal{Q}}$ | Original and quantized bit-widths, respectively |
| $\delta_f(\cdot)$ | Error induced by applying $f(\cdot)$ |
| $t \in \mathcal{T}_\phi$ | Abstract data type from the set of all valid types in model $\phi$ |
| $t_f$ | Granularity of $f(\cdot)$ |
| $u \in \mathbb{U}(\phi; t)$ | A unit of type $t$ within the model $\phi$ |
| $\mathbb{D}_u^f(\pi)$ | Binary indicator of whether $f(\cdot)$ modifies unit $u$ under order $\pi$ |
| $\mathcal{G}(f_1, f_2)$ | Performance gap between $f_1(\cdot)$ and $f_2(\cdot)$ |
| $\mathcal{A}(f_1 \to f_2)$ | Compression order advantage of $f_1 \to f_2$ over $f_2 \to f_1$ |
| $C_f^*$ | Compression Equivalent Ratio (CER) of $f(\cdot)$ |
| $\Delta(\phi; f_1 \to f_2)$ | Interference from $f_1(\cdot)$ to $f_2(\cdot)$ |

## B  Details on Theoretical Analysis

We provide the detailed proofs for Theorems 1 and 2, then formulate a generalized version of Hypothesis 1 applicable to a broader setting with multiple compression methods.

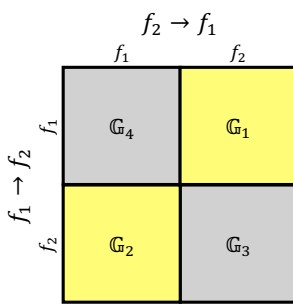

Figure 11: We partition all units $u$ into four disjoint groups. Only groups $\mathbb{G}_1$ and $\mathbb{G}_2$ influence $\mathcal{A}(f_1 \to f_2)$.

### B.1  Proof of Theorem 1

*Proof.* Given two compression methods $f_1(\cdot)$ and $f_2(\cdot)$ with respective granularities $t_{f_1}$ and $t_{f_2}$, disjoint selectivity ensures that every unit is assigned exclusively to one method. Hence, every unit $u \in \mathbb{U}(\phi; t_{\text{lut}}(t_{f_1}, t_{f_2}))$ is classified into one of four disjoint groups, $\mathbb{G}_1, \mathbb{G}_2, \mathbb{G}_3$, or $\mathbb{G}_4$, according to its assigned method. Then,

$$\mathbb{G}_1 = \{u \mid \mathbb{D}_u^{f_1}(f_2 \circ f_1) = 1, \ \mathbb{D}_u^{f_1}(f_1 \circ f_2) = 0\},$$
$$\mathbb{G}_2 = \{u \mid \mathbb{D}_u^{f_1}(f_2 \circ f_1) = 0, \ \mathbb{D}_u^{f_1}(f_1 \circ f_2) = 1\},$$
$$\mathbb{G}_3 = \{u \mid \mathbb{D}_u^{f_1}(f_2 \circ f_1) = 0, \ \mathbb{D}_u^{f_1}(f_1 \circ f_2) = 0\},$$
$$\mathbb{G}_4 = \{u \mid \mathbb{D}_u^{f_1}(f_2 \circ f_1) = 1, \ \mathbb{D}_u^{f_1}(f_1 \circ f_2) = 1\},$$
$$\mathbb{G}_1 \cup \mathbb{G}_2 \cup \mathbb{G}_3 \cup \mathbb{G}_4 = \mathbb{U}(\phi; t_{\text{lut}}(t_{f_1}, t_{f_2})),$$

where $\mathbb{U}(\phi; t)$ represents the unit set of model $\phi$ at granularity $t$, and $\mathbb{D}_u^f(\pi)$ records whether $f(\cdot)$ modifies $u$ under the ordering $\pi$ (1 if yes, 0 if no). Note that these four groups are mutually exclusive and collectively exhaustive. Also, $|\mathbb{G}_1| = |\mathbb{G}_2|$ since compression ratios $C_{f_1}$ and $C_{f_2}$ are identical regardless of the compression order. Figure 11 illustrates the four groups.

Under Assumption 1 and the defined partitioning of groups, the compression order advantage $\mathcal{A}(\,\cdot\,)$ is expressed in terms of unit-wise reconstruction errors $\delta_f(u_i)$:

$$
\begin{aligned}
&\mathcal{A}\big(f_1 \to f_2\big) \\
&= \mathcal{M}((f_2 \circ f_1)(\phi)) - \mathcal{M}((f_1 \circ f_2)(\phi)) \\
&= -\beta\big(\delta_{(f_2 \circ f_1)}(\phi) - \delta_{(f_1 \circ f_2)}(\phi)\big) \\
&= -\beta\Big( \sum_{u_i \in \mathbb{G}_1} \|\delta_{f_1}(u_i)\|_F^2 + \sum_{u_i \in \mathbb{G}_2} \|\delta_{f_2}(u_i)\|_F^2 + \sum_{u_i \in \mathbb{G}_3} \|\delta_{f_2}(u_i)\|_F^2 + \sum_{u_i \in \mathbb{G}_4} \|\delta_{f_1}(u_i)\|_F^2 \\
&\qquad\quad - \sum_{u_i \in \mathbb{G}_1} \|\delta_{f_2}(u_i)\|_F^2 - \sum_{u_i \in \mathbb{G}_2} \|\delta_{f_1}(u_i)\|_F^2 - \sum_{u_i \in \mathbb{G}_3} \|\delta_{f_2}(u_i)\|_F^2 - \sum_{u_i \in \mathbb{G}_4} \|\delta_{f_1}(u_i)\|_F^2 \Big) \\
&= -\beta\Big( \sum_{u_i \in \mathbb{G}_1} \|\delta_{f_1}(u_i)\|_F^2 + \sum_{u_i \in \mathbb{G}_2} \|\delta_{f_2}(u_i)\|_F^2 - \sum_{u_i \in \mathbb{G}_1} \|\delta_{f_2}(u_i)\|_F^2 - \sum_{u_i \in \mathbb{G}_2} \|\delta_{f_1}(u_i)\|_F^2 \Big) \\
&= \beta\Big( \sum_{u_i \in \mathbb{G}_2} g(u_i) - \sum_{u_i \in \mathbb{G}_1} g(u_i)\Big),
\end{aligned}
$$

where error difference $g(u_i) = \big\|\delta_{f_1}(u_i)\big\|_F^2 - \big\|\delta_{f_2}(u_i)\big\|_F^2$. Note that $\mathbb{G}_3$ and $\mathbb{G}_4$ are discarded since their effect remains unchanged irrespective of the compression order.

$\square$

**Case study on pruning and quantization.** To support intuition, we provide a case study on pruning and quantization, which constitute the core scenario of our work. As described in the main text, disjoint selectivity holds only when the granularity $t_{\mathcal{P}}$ of pruning $\mathcal{P}(\cdot)$ is greater than or equal to the granularity $t_{\mathcal{Q}}$ of quantization $\mathcal{Q}(\cdot)$. We analyze this at the layer level: let $\mathbf{W}_i$ and $\mathbf{X}_i$ denote the weight and activation of a layer $l_i \in \mathbb{L}$ in the model $\phi$. Note that the error $\delta_f(\mathbf{W}_i, \mathbf{X}_i) = f(\mathbf{W}_i)f(\mathbf{X}_i) - \mathbf{W}_i\mathbf{X}_i$ for a compression method $f(\cdot)$, as described in Section 2.

We partition the layers $\mathbb{L}$ into four disjoint groups $\mathbb{G}_1$, $\mathbb{G}_2$, $\mathbb{G}_3$, and $\mathbb{G}_4$ based on their pruning status:

$$
\begin{aligned}
\mathbb{G}_1 &= \{u \mid \mathbb{D}_u^{\mathcal{Q}}(\mathcal{P} \circ \mathcal{Q}) = 1,\ \mathbb{D}_u^{\mathcal{Q}}(\mathcal{Q} \circ \mathcal{P}) = 0\}, \\
\mathbb{G}_2 &= \{u \mid \mathbb{D}_u^{\mathcal{Q}}(\mathcal{P} \circ \mathcal{Q}) = 0,\ \mathbb{D}_u^{\mathcal{Q}}(\mathcal{Q} \circ \mathcal{P}) = 1\}, \\
\mathbb{G}_3 &= \{u \mid \mathbb{D}_u^{\mathcal{Q}}(\mathcal{P} \circ \mathcal{Q}) = 0,\ \mathbb{D}_u^{\mathcal{Q}}(\mathcal{Q} \circ \mathcal{P}) = 0\}, \\
\mathbb{G}_4 &= \{u \mid \mathbb{D}_u^{\mathcal{Q}}(\mathcal{P} \circ \mathcal{Q}) = 1,\ \mathbb{D}_u^{\mathcal{Q}}(\mathcal{Q} \circ \mathcal{P}) = 1\}, \\
\mathbb{G}_1 &\cup \mathbb{G}_2 \cup \mathbb{G}_3 \cup \mathbb{G}_4 = \mathbb{L},
\end{aligned}
$$

where $\mathbb{D}_u^f(\pi)$ records whether $f(\cdot)$ modifies $u$ under the ordering $\pi$ (1 if yes, 0 if no). In the pruning–quantization setting, the partition is directly determined by whether each layer is pruned in the final model. Pruning behaves as an absorbing operator: pruning overrides any modification introduced by quantization. Therefore, the partition above reduces to grouping layers according to whether they are pruned under $\mathcal{Q} \circ \mathcal{P}$ or $\mathcal{P} \circ \mathcal{Q}$, yielding the pruning-status-based formulation below:

$$
\begin{aligned}
\mathbb{G}_1 &= \mathbb{P}_{\mathcal{Q} \circ \mathcal{P}} \setminus \mathbb{P}_{\mathcal{P} \circ \mathcal{Q}}, \\
\mathbb{G}_2 &= \mathbb{P}_{\mathcal{P} \circ \mathcal{Q}} \setminus \mathbb{P}_{\mathcal{Q} \circ \mathcal{P}}, \\
\mathbb{G}_3 &= \mathbb{P}_{\mathcal{Q} \circ \mathcal{P}} \cap \mathbb{P}_{\mathcal{P} \circ \mathcal{Q}}, \\
\mathbb{G}_4 &= \mathbb{L} \setminus (\mathbb{P}_{\mathcal{Q} \circ \mathcal{P}} \cup \mathbb{P}_{\mathcal{P} \circ \mathcal{Q}}), \\
\mathbb{G}_1 &\cup \mathbb{G}_2 \cup \mathbb{G}_3 \cup \mathbb{G}_4 = \mathbb{L},
\end{aligned}
$$

where $\mathbb{P}_f$ denote the sets of pruned layers when applying $f(\cdot)$.

Then, the quantization-first advantage $\mathcal{A}(\mathcal{Q} \to \mathcal{P})$ is estimated as follows:

$$
\begin{aligned}
\mathcal{A}(\mathcal{Q} \to \mathcal{P}) &= \mathcal{M}((\mathcal{P} \circ \mathcal{Q})(\phi)) - \mathcal{M}((\mathcal{Q} \circ \mathcal{P})(\phi)) = -\beta\big(\delta_{\mathcal{P} \circ \mathcal{Q}}(\phi) - \delta_{\mathcal{Q} \circ \mathcal{P}}(\phi)\big) \\
&= -\beta\Big( \sum_{l_i \in \mathbb{L}} \big\| \delta_{\mathcal{P} \circ \mathcal{Q}}(\mathbf{W}_i, \mathbf{X}_i) \big\|_F^2 - \big\| \delta_{\mathcal{Q} \circ \mathcal{P}}(\mathbf{W}_i, \mathbf{X}_i) \big\|_F^2 \Big) \\
&= -\beta\Big( \sum_{l_i \in \mathbb{G}_1} \big\{ \big\| \delta_{\mathcal{Q}}(\mathbf{W}_i, \mathbf{X}_i) \big\|_F^2 - \| -\mathbf{W}_i\mathbf{X}_i \|_F^2 \big\} + \sum_{l_i \in \mathbb{G}_2} \big\{ \| -\mathbf{W}_i\mathbf{X}_i \|_F^2 - \big\| \delta_{\mathcal{Q}}(\mathbf{W}_i, \mathbf{X}_i) \big\|_F^2 \big\} \Big) \\
&= \beta\Big( \sum_{l_i \in \mathbb{G}_2} g(\mathbf{W}_i, \mathbf{X}_i) - \sum_{l_i \in \mathbb{G}_1} g(\mathbf{W}_i, \mathbf{X}_i) \Big),
\end{aligned}
$$

where $g(\mathbf{W}_i, \mathbf{X}_i) = \big\| \delta_{\mathcal{Q}}(\mathbf{W}_i, \mathbf{X}_i) \big\|_F^2 - \| -\mathbf{W}_i\mathbf{X}_i \|_F^2$. This expression holds as for any layer $l_i \in \mathbb{L}$, the pruning operator and its associated error are defined as follows:

$$
\mathcal{P}(\mathbf{W}_i)\mathcal{P}(\mathbf{X}_i) = \begin{cases} \mathbf{0} & \text{if pruned} \\ \mathbf{W}_i\mathbf{X}_i & \text{otherwise} \end{cases}, \quad \delta_{\mathcal{P}}(\mathbf{W}_i, \mathbf{X}_i) = \begin{cases} -\mathbf{W}_i\mathbf{X}_i & \text{if pruned} \\ \mathbf{0} & \text{otherwise} \end{cases}.
$$

### B.2 Proof of Theorem 2

*Proof.* As $\mathcal{A}(\mathcal{Q} \to \mathcal{P})$ and $\mathcal{G}(\cdot)$ (or $C_{\mathcal{P}}^* - C_{\mathcal{Q}}$) are functions of the compression ratio $C_{\mathcal{Q}}$ we analyze their behavior separately. Without loss of generality, we consider only the case where $C_{\mathcal{Q}}$ changes in the direction of increasing $C_{\mathcal{P}}^* - C_{\mathcal{Q}}$, i.e., decreasing $C_{\mathcal{Q}}$.

Under Assumption 2, which assumes well-designed quantization, decreasing $C_{\mathcal{Q}}$ preserves the expected value of the quantized outputs while decreasing their standard deviation. As pruning intensity is held constant, the variation across compression orders is attributed solely to the severity of quantization. Lower quantization ratio (i.e., smaller standard deviation) decreases the chance that units behave differently across orders, which can only decrease or preserve the value of $|\mathbb{G}_1| = |\mathbb{G}_2|$, but never increase it. We analyze the two possible cases as follows.

- **Case 1: Number of layers affected by order decreases.** Although more than one layer may change simultaneously, any such change can be decomposed into a sequence in which layers are added one by one; thus it suffices to analyze the case where exactly one layer is added. Let $l_i$ and $l_j$ denote the layers moving from $\mathbb{G}_1$ and $\mathbb{G}_2$ respectively. Such a transition occurs in a budget-preserving manner: one layer from $\mathbb{G}_1$ and one from $\mathbb{G}_2$ are jointly reallocated, with one moving to $\mathbb{G}_3$ and the other to $\mathbb{G}_4$, so that the number of pruned layers remains fixed. As only $\mathbb{G}_1$ and $\mathbb{G}_2$ contribute to $\mathcal{A}(\mathcal{Q} \to \mathcal{P})$, removing $l_i \in \mathbb{G}_1$ and $l_j \in \mathbb{G}_2$ from these groups changes the value by $-\beta\big(g(l_j) - g(l_i)\big)$ (see Theorem 1). Therefore, to show that $\mathcal{A}(\mathcal{Q} \to \mathcal{P})$ does not decrease, it suffices to prove that $g(l_j) - g(l_i) \le 0$. Expanding the definition of $g(\cdot)$, we obtain

$$
g(l_j) - g(l_i) = \big( \|\delta_{\mathcal{Q}}(\mathbf{W}_j, \mathbf{X}_j)\|_F^2 - \|\delta_{\mathcal{Q}}(\mathbf{W}_i, \mathbf{X}_i)\|_F^2 \big) - \big( \| -\mathbf{W}_j\mathbf{X}_j \|_F^2 - \| -\mathbf{W}_i\mathbf{X}_i \|_F^2 \big).
$$

  Under Assumptions 1 and 2, the second term in parentheses is positive, i.e., $\| -\mathbf{W}_j\mathbf{X}_j \|_F^2 - \| -\mathbf{W}_i\mathbf{X}_i \|_F^2 > 0$. This is because under pruning alone, $l_i \in \mathbb{G}_1$ is pruned while $l_j \in \mathbb{G}_2$ is not. Under the well-designed pruning assumption, which minimizes performance drop, this is equivalent to minimizing the increased error. Therefore, the pruning error $| -\mathbf{W}_i\mathbf{X}_i |_F^2$ for pruned $l_i$ is less than or equal to $| -\mathbf{W}_j\mathbf{X}_j |_F^2$ for unpruned $l_j$, making the term positive.

  Given the assumption of well-designed quantization in Assumption 2, the remaining first term, which denotes the difference in quantization errors, is negligible compared to the pruning-related component. This is because the quantization error at each layer is modeled as zero-mean noise with small variance, whereas the pruning error term $\| -\mathbf{W}_i\mathbf{X}_i \|_F^2$ (or $\| -\mathbf{W}_j\mathbf{X}_j \|_F^2$) corresponds directly to the magnitude of the pruned responses. Thus, the difference $\|\delta_{\mathcal{Q}}(\mathbf{W}_j, \mathbf{X}_j)\|_F^2 - \|\delta_{\mathcal{Q}}(\mathbf{W}_i, \mathbf{X}_i)\|_F^2$ remains uniformly small compared to $\| -\mathbf{W}_j\mathbf{X}_j \|_F^2 - \| -\mathbf{W}_i\mathbf{X}_i \|_F^2$, so the pruning-induced gap dominates $g(l_j) - g(l_i)$. Consequently, we get

$$
g(l_j) - g(l_i) \approx -\big( \| -\mathbf{W}_j\mathbf{X}_j \|_F^2 - \| -\mathbf{W}_i\mathbf{X}_i \|_F^2 \big) < 0.
$$

  Overall, the decrease in the number of order-dependent layers leads to the increase of the order advantage $\mathcal{A}(\mathcal{Q} \to \mathcal{P})$.

Table 3: Baseline methods covered in our experiments across different settings.

| Compression | Modality | Target Models | Baseline Methods |
|---|---|---|---|
| Pruning $\mathcal{P}(\cdot)$ | Language models | Decoder-only models | SparseGPT (Frantar & Alistarh, 2023), Wanda (Sun et al., 2024), SLEB (Song et al., 2024) |
| | | Encoder-based models | K-prune (Park et al., 2024) |
| | Vision models | CNNs | PRACTISE (Wang & Wu, 2023) |
| | | ViTs | SAViT (Chuanyang et al., 2022) |
| Quantization $\mathcal{Q}(\cdot)$ | Language models | Decoder-only models | RTN (Gupta et al., 2015), OPTQ (Frantar et al., 2023), QuaRoT (Ashkboos et al., 2024b) |
| | | Encoder-based models | UniQuanF (Park et al., 2025a) |
| | Vision models | CNNs | N2UQ (Liu et al., 2022) |
| | | ViTs | RepQ-ViT (Li et al., 2023) |
| Parameter Efficient Fine-tuning | Language models | Decoder-only models | LoRA (Hu et al., 2022) |
| Parameter Sharing | Language models | Decoder-only models | Basis Sharing (Wang et al., 2025a) |
| Mixed-precision quantization | Vision models | CNNs | HAWQ-V2 (Dong et al., 2020) |

- **Case 2: Number of layers affected by order remains unchanged.** As the loss-contributing groups $\mathbb{G}_1$ and $\mathbb{G}_2$ do not change, the compression order advantage $\mathcal{A}(\mathcal{Q} \to \mathcal{P})$ remains unaffected.

Therefore, in all cases where $C_{\mathcal{Q}}$ decreases, the value of $\mathcal{A}(\mathcal{Q} \to \mathcal{P})$ does not decrease. In conclusion, monotonicity between $\mathcal{A}(\mathcal{Q} \to \mathcal{P})$ and $\mathcal{G}(\cdot)$ holds under fixed $C_{\mathcal{P}}$. $\qquad\square$

### B.3 GENERALIZATION TO MULTIPLE METHODS

We formulate Hypothesis 1 with two compression methods $f_1(\cdot)$ and $f_2(\cdot)$. This is because if the hypothesis holds for any pair of methods, it can be generalized to more than two methods.

Following the setup in Problem 1, suppose we sequentially apply a set $\mathbb{F} = f_1(\cdot), f_2(\cdot), \cdots, f_n(\cdot)$ of compression methods to a pre-trained model $\phi$. Then, any pair $(\pi_1, \pi_2)$ of permutations from the set $\Pi = \{\pi : \mathbb{F} \to \mathbb{F} \mid \pi \text{ is bijective}\}$ of all permutations can be converted into one another via a sequence of adjacent transpositions. This is because the adjacent transpositions generate the full symmetric group, allowing any permutation to be constructed from another. Thus, under Hypothesis 1, our original claim shown in Figure 1 holds, that applying stronger permutations later leads to better performance of the compressed model.

## C EXPERIMENTAL SETUP

We describe the details on the experimental setup, including models, datasets, baselines, evaluation protocol, and implementation.

**Models.** We evaluate representative models across modalities, including LLaMA 2 (7B, 13B) (Touvron et al., 2023), LLaMA 3 8B (Grattafiori et al., 2024), Mistral 7B (Jiang et al., 2023), Mistral Nemo-12B (Mistral AI Team, 2024), and BERT (Devlin et al., 2019) for language, and ResNet-18 (He et al., 2016) and DeiT-Base (Touvron et al., 2021) for vision.

**Datasets.** We evaluate decoder-only language models on WikiText-2 (Merity et al., 2017) and C4 (Raffel et al., 2020) datasets for perplexity, and on five commonsense reasoning tasks—ARC (Clark et al., 2018), HellaSwag (Zellers et al., 2019), LAMBADA (Paperno et al., 2016), PIQA (Bisk et al., 2020), and Winogrande (Sakaguchi et al., 2021). For encoder-based models, we evaluate the performance using Spearman's rank correlation coefficient on the STS-B dataset. For vision models, we report classification accuracy on ImageNet (ILSVRC 2012) (Deng et al., 2009) dataset.

**Baseline Methods.** We validate our hypothesis across a total of sixteen pairs of compression methods by incorporating six pruning methods, six quantization methods, and one mixed-precision quantization method. Table 3 provides an overview of baseline methods categorized by target models and modalities. Refer to the original papers for further details.

**Evaluation Protocol.** The calibration dataset consists of a single batch with 128 samples drawn from the same dataset used for perplexity evaluation. We set the batch size to 16 for perplexity evaluation and to 128 for commonsense reasoning tasks. All quantization methods apply the same bit-width to weights, activations, and KV-cache, with clipping applied during weight quantization. Both models are evaluated without fine-tuning using a batch size of 128. Metrics are reported as the average of five repeated runs, each computed with four-digit precision. We plot the relative values for visualization.

**Implementation and Machine.** Our implementations are written in Python and rely on PyTorch, Transformers, Accelerate, and TorchVision libraries. For all baseline methods, we reproduce the results based on their open-source code and hyperparameter configurations. All of our experiments are done at a workstation with Intel Xeon Gold 6338 and NVIDIA A100 80GB.

**Parameter Efficient Fine-tuning Experiment.** We adopt LoRA (Hu et al., 2022) on top of SparseGPT (Frantar & Alistarh, 2023) and RTN (Gupta et al., 2015) to fine-tune the compressed model. The target model is LLaMA 3 8B (Grattafiori et al., 2024), where fine-tuning is processed after quantization. We select WikiText-2 as the calibration dataset and train for a total of 2 epochs. We follow Basis Sharing (Wang et al., 2025a) for training details of the low-rank adapter, while exploiting the PEFT (Mangrulkar et al., 2022) library.

**Parameter Sharing Experiment.** We evaluate the performance of LLaMA 2 7B model when applying Basis Sharing (Wang et al., 2025a) and magnitude-based pruning (Han et al., 2015). We follow Basis Sharing for hyperparameters regarding parameter sharing, where the group size is 128. QuaRot (Ashkboos et al., 2024b) is selected as the quantization baseline for calculating compression equivalent bits. No additional fine-tuning is applied under this setting.

**Mixed-precision Quantization Experiment.** We base our method on HAWQ-V2 (Dong et al., 2020), but allocate bit-widths iteratively rather than in a single shot following LampQ (Kim et al., 2026). At each iteration, we search per-layer bit-widths from a range of [2, 3, 4, 5, 6, 7, 8] and train for 5 epochs. All other experimental settings, including hyperparameters and quantization techniques, are aligned with the original paper. We run all MPQ experiments on a workstation with Intel Xeon Silver 4310 and NVIDIA RTX 4090.

# D  FURTHER DISCUSSION AND EXPERIMENTS

We present results from extended experiments, and offer further discussion and remarks on our work.

## D.1  EXPERIMENTS ON DIVERSE LLMS

Our experiments on decoder-only models are limited to LLaMA herd models (LLaMA 2 (Touvron et al., 2023) and LLaMA 3 (Grattafiori et al., 2024) models), which may not fully reflect broader generality. To address this, we conduct additional experiments on models from the Mistral herd. Figure 12 presents the results of applying SparseGPT ($\mathcal{P}(\cdot)$) and QuaRot ($\mathcal{Q}(\cdot)$) to Mistral 7B (Jiang et al., 2023) and Mistral Nemo 12B (Mistral AI Team, 2024). We have two observations from the result. First, the compression-order trend aligns well with the hypothesis across Mistral-based models. The result serves as additional evidence confirming the hypothesis in decoder-only language models. Second, comparing models within the same herd (also refer to Figure 3), we find that smaller models exhibit greater variation in compression-order advantage for identical CER differences. This may be because low-bit quantization (or stronger quantization) causes greater degradation in smaller models, thereby intensifying the observed differences.

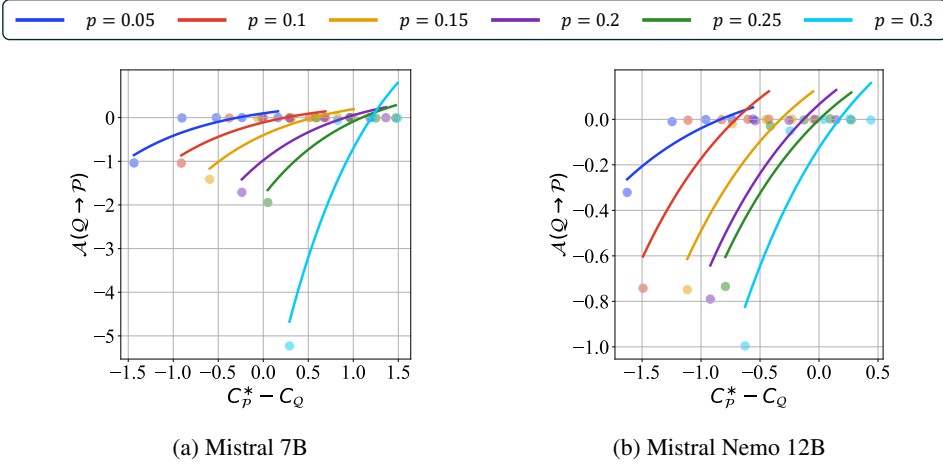

(a) Mistral 7B          (b) Mistral Nemo 12B

Figure 12: The compression order advantage $\mathcal{A}(\mathcal{Q} \to \mathcal{P})$ increases monotonically with the CER difference $C_{\mathcal{P}}^* - C_{\mathcal{Q}}$ also for Mistral herd models. See Appendix D.1 for details.

### D.2 ANALYSIS ON ENCODER-BASED MODELS

Beyond decoder-only LLMs, we extend our analysis to encoder-based language models to validate the generality of our hypothesis. Figure 13 presents the performance of a BERT (Devlin et al., 2019) model under K-prune (Park et al., 2024) ($\mathcal{P}(\cdot)$) and Uni-QuanF (Park et al., 2025a) ($\mathcal{Q}(\cdot)$). We adopt Spearman correlation as the performance metric, measured on STS-B dataset (Cer et al., 2017) from GLUE (Wang et al., 2019) benchmark. We observe a monotonic increase along both axes of $\mathcal{A}(\mathcal{Q} \to \mathcal{P})$ and $C_{\mathcal{P}}^* - C_{\mathcal{Q}}$, confirming that our hypothesis holds.

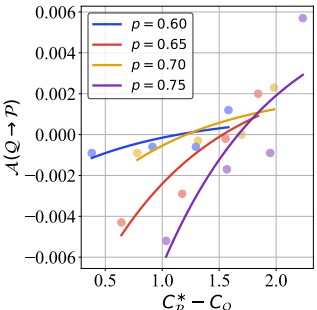

Figure 13: The hypothesis holds for encoder-based language models. See Appendix D.2 for details.

### D.3 COMMONSENSE REASONING PERFORMANCE

Although the negative of perplexity serves as an intuitive and efficient metric $\mathcal{M}(\cdot)$ for evaluating language models, prior studies (Meister & Cotterell, 2021; Fang et al., 2025) suggest it does not always correlate with real-world performance. We thus investigate the performance of LLaMA 3 8B model across five commonsense reasoning tasks in Figure 14, when applying SparseGPT ($\mathcal{P}(\cdot)$) and QuaRot ($\mathcal{Q}(\cdot)$). Results affirm the generality and metric-agnostic nature of our framework, as the hypothesis holds across these tasks.

### D.4 IMPACT OF ROTATION ON PRUNING METHODS

In Figure 5 and Finding 3, we observe that applying rotation without quantization may lead to notable degradation on pruning performance. To further analyze this, Table 4 compares the performance of LLaMA 3 8B model pruned with and without QuaRot-based rotation, across two pruning methods with different granularities. We have two observations from the result. First, rotation-induced degradation scales with the pruning ratio. This is because higher pruning ratios result in more units being pruned that are altered by rotation, thereby increasing the error. Second, unstructured pruning (SparseGPT) exhibits significantly higher error compared to structured pruning (SLEB). This trend is especially evident under high pruning ratios.

We therefore investigate the underlying reason behind this phenomenon. We identify two types of errors induced by pruning, depending on its granularity: matrix-wise and element-wise errors. Figure 15 conceptually illustrates these two cases.

First, in the case of matrix-wise pruning, ignoring rotation during pruning leaves the rotation-induced matrix $\mathbf{H}$ intact, introducing extra computation and numerical errors compared to the non-rotated case. As suggested in QuaRot (Ashkboos et al., 2024b), the rotation inverse is fused into the target layer,

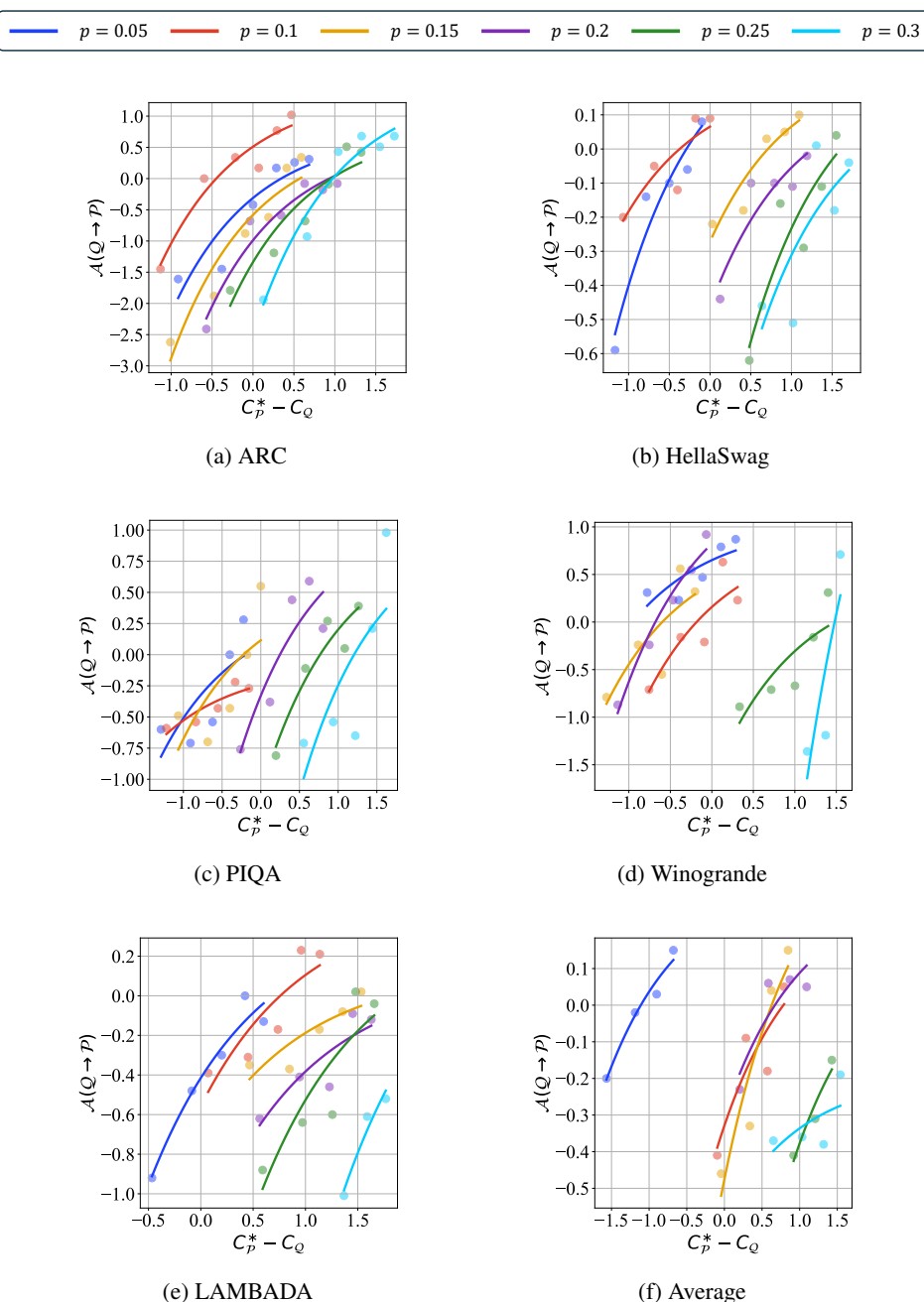

Figure 14: Commonsense reasoning task performance of a LLaMA 3 8B model for SparseGPT and QuaRot. See Appendix D.3 for details.

Table 4: WikiText-2 perplexity comparison of a LLaMA 3 8B model pruned by SLEB (Song et al., 2024) and SparseGPT (Frantar & Alistarh, 2023) under varying pruning ratios, with and without rotation following QuaRot (Ashkboos et al., 2024b). See Section D.4 for details.

| Pruning Ratio | SLEB | | | SparseGPT | | |
|---|---|---|---|---|---|---|
| | No rotation | Rotation | **Difference** | No rotation | Rotation | **Difference** |
| Original | | | 6.137 | | | |
| 0.05 | 6.857 | 6.871 | **0.014** | 6.140 | 6.154 | **0.014** |
| 0.1 | 8.792 | 8.828 | **0.036** | 6.159 | 6.205 | **0.046** |
| 0.15 | 12.603 | 12.615 | **0.012** | 6.213 | 6.352 | **0.139** |
| 0.2 | 25.289 | 25.295 | **0.006** | 6.330 | 6.629 | **0.299** |
| 0.25 | 51.212 | 51.560 | **0.348** | 6.546 | 7.250 | **0.704** |
| 0.3 | 61.502 | 61.901 | **0.399** | 6.894 | 8.504 | **1.610** |
| 0.35 | 65.997 | 66.234 | **0.237** | 7.474 | 20.842 | **13.368** |
| 0.4 | 92.848 | 93.260 | **0.412** | 8.477 | 98.213 | **89.736** |

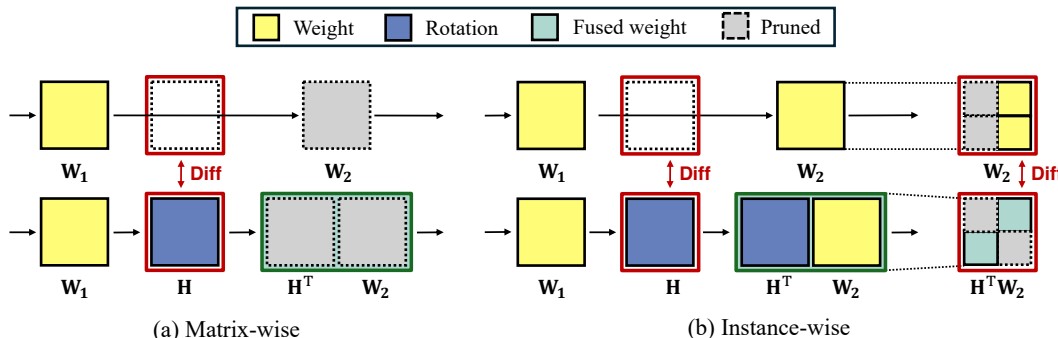

Figure 15: Two cases of errors when pruning rotated units. See Section D.4 for details.

while the original transform is merged into the preceding normalization layer, leaving un-removed components that generate error during naïve pruning. This type of error scales proportionally with the pruning ratio, as each pruned matrix introduces one such error.

Second, in element-wise pruning, additional errors arise due to rotation-induced changes in unit selection, on top of the matrix-wise error. As the goal of rotation is to facilitate quantization by flattening activation outliers, multiplying its inverse results in an error compared to the original matrix. Consequently, the discrepancy in layer content leads to different pruning decisions. Furthermore, this selection-based error grows with higher pruning ratios due to a greater number of pruned units.

In summary, given these errors, it is crucial to develop pruning techniques that align with rotation-based quantization strategies.

### D.5 A DIRECT COMPARISON WITH PRIOR STUDIES

We discuss how our approach differs from prior studies, particularly SmoothQuant (Xiao et al., 2023) and Harma et al. (2025). Our contribution lies in establishing a general analysis for understanding compression order across diverse methods, whereas these prior works either focus on single-method optimization or analyze specific method pairs under restrictive assumptions.

**SmoothQuant (Xiao et al., 2023).** SmoothQuant addresses a fundamentally different problem than joint compression order optimization. Specifically, SmoothQuant optimizes a *single* compression technique (quantization) by mitigating activation outliers through per-channel scaling transformations. While the paper states that "SmoothQuant is orthogonal to quantization schemes," this refers to its compatibility as a pre-processing step that can be applied before various quantization methods. However, SmoothQuant does not examine the order-dependent interaction problem when combining

quantization with other compression families such as pruning. In contrast, our work focuses on understanding how compression order affects the model performance when sequentially combining methods from different compression families. SmoothQuant may serve as a component within our quantization baselines (i.e., as a pre-processing step before quantization), but our analysis operates at a higher level—determining the optimal ordering between different model compression techniques regardless of the specific quantization implementation. This distinction is critical: SmoothQuant addresses *intra-method* optimization (improving quantization itself), while we address *inter-method* composition (ordering across different compression types).

**Harma et al. (2025).** As noted in Section 2, only a few studies have addressed how the order of compression methods affects the model performance. Among them, Harma et al. (2025) stands out as the only study that attempts a theoretical approach to the problem. They examine the interaction between pruning and quantization, showing that the two are not orthogonal as assumed by previous works. Furthermore, they argue that pruning followed by quantization is universally optimal.

However, their framework suffers from three significant limitations. First, their framework relies on oversimplified assumptions that hinder practical applicability. Specifically, they focus solely on magnitude-based pruning (removes weights based on their absolute values) and max-scaled block-wise quantization (uniformly rescaling blocks using their maximum value), both of which are naïve approaches that are less practical and often fail to preserve accuracy. Second, their analysis is confined to a minimal set of scenarios, failing to address diverse architectures or methods. Beyond the limited set of methods, their experiments also consider only the combination of two techniques—pruning and quantization—on decoder-based LLMs, lacking broader coverage of models and compression approaches. Lastly, the framework cannot be generalized across different settings, as many counterexamples have shown that pruning-before-quantization is not always optimal. Motivated by these gaps, we aim for a more general formulation that holds across methods, models, and metrics, thereby introducing the Progressive Intensity Hypothesis.

### D.6 VIOLATION CASES OF THE HYPOTHESIS

Although our hypothesis is highly general and robust, we still observe cases where it does not hold. These cases largely fall into three categories: severe performance collapse, full model re-training, and increase of order-affected layers. First, each model exhibits a different tolerance to compression, with performance dropping exponentially beyond a certain ratio. While these settings are impractical due to severe performance loss, we observe cases where applying the stronger method first performs better. This may be because the error is already too large, violating our assumption of well-designed compression in Section 4; applying the stronger method first might help reduce the total error. For less compression-robust models like decoder-based LLMs, we observe earlier breakdowns—such as diminishing advantage when pruning ratio increases at fixed bit-width (Figure 3a). Second, when strong full-training is applied, the advantage from compression order may invert. Compression order serves merely as initialization, and the retraining process dominates, making it difficult to attribute outcomes to order alone. We plan to investigate these and potentially other exceptions more rigorously in future work. Lastly, in practical situations, increasing $C_{\mathcal{P}}$ may result in increase of order-affected layers, leading to a violation of the hypothesis. We analyze the details in the following paragraph.

**Impact of $C_{\mathcal{P}}$.** Without loss of generality, we consider the case when only $C_{\mathcal{P}}$ increases. Increasing $C_{\mathcal{P}}$ implies a stronger pruning effect, since it lowers $\mathcal{M}(\mathcal{P}(\phi))$, resulting in a decrease in $\mathcal{G}(\mathcal{P}, \mathcal{Q})$ and a corresponding increase in $C_{\mathcal{P}}^*$. Hence, to ensure monotonicity and satisfy Hypothesis 1, $\mathcal{A}(\mathcal{Q} \to \mathcal{P})$ should increase accordingly. Note that $C_{\mathcal{Q}}$ is fixed while analyzing the effect of $C_{\mathcal{P}}$.

To analyze the effect of increasing $C_{\mathcal{P}}$, we first consider a local step in which the total number of pruned units increases by one. For the initial pruning ratio $p$ and the increased ratio $p'$ under the same granularity $t_{\mathcal{P}}$, the following relation holds:

$$p' \cdot |\mathbb{U}(\phi; t_{\mathcal{P}})| = p \cdot |\mathbb{U}(\phi; t_{\mathcal{P}})| + 1.$$

From the definition of compression ratio $C_{\mathcal{P}} = 1/(1-p)$, larger pruning ratios correspond to larger compression ratios. By repeating this incremental process, we can construct any pruning ratio. Under disjoint selectivity, each unit is exclusively assigned to one compression method, allowing us to partition the units into four disjoint groups as discussed in Appendix B.1.

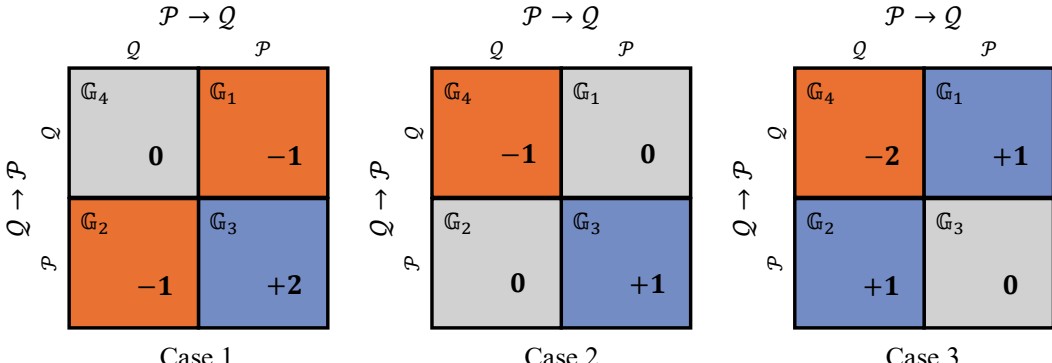

Figure 16: If the number of pruned units increases by one, the change in unit allocation across groups fall into three distinct cases. See Appendix D.6 for details.

Increasing the pruning ratio from $p$ to $p'$ result in three possible changes in the group configuration: the number of affected layers by order (i.e., $|\mathbb{G}_1|$ or $|\mathbb{G}_2|$; note that two values are equal.) 1) decreases by one, 2) remains unchanged, or 3) increases by one. The three cases are visualized in Figure 16. From Theorem 1, $\mathcal{A}(\mathcal{Q} \to \mathcal{P}) = \beta \cdot \left( \sum_{l_i \in \mathbb{G}_2} g(l_i) - \sum_{l_i \in \mathbb{G}_1} g(l_i) \right)$ (where $g(l_i) = \|\delta_{\mathcal{Q}}(\mathbf{W}_i, \mathbf{X}_i)\|_F^2 - \| - \mathbf{W}_i\mathbf{X}_i\|_F^2$) should be preserved or increased to satisfy Hypothesis 1. However, this condition is fulfilled in only Cases 1 and 2, but not in Case 3.

- **Case 1: Number of layers affected by order decreases by one.** If a layer is no longer affected by compression due to order change, then another layer must also be excluded to preserve the total number of pruned layers which should be increased by one. Thus, $|\mathbb{G}_1|$ and $|\mathbb{G}_2|$ each decrease by one, while $|\mathbb{G}_3|$ increases by two. Similar to Case 1 of Appendix B.2, as the increase in $C_{\mathcal{P}}$ eliminates a negative loss term, $\mathcal{A}(\mathcal{Q} \to \mathcal{P})$ increases.

- **Case 2: Number of layers affected by order remains unchanged.** If the additionally pruned layer is always pruned regardless of the compression order, then $|\mathbb{G}_3|$ increases by 1 while $|\mathbb{G}_4|$ decreases by 1. Similar to Case 2 of Appendix B.2, as the loss-contributing groups $\mathbb{G}_1$ and $\mathbb{G}_2$ do not change, the compression order advantage $\mathcal{A}(\mathcal{Q} \to \mathcal{P})$ remains unaffected.

- **Case 3: Number of layers affected by order increases by one.** The aforementioned two cases could not increase the number of order-dependent units, whereas there should exist a case where both groups $|\mathbb{G}_1|$ and $|\mathbb{G}_2|$ increases by one. As the number of total pruned layers should be increased by one, the added layers should be originated from $\mathbb{G}_4$, i.e., $|\mathbb{G}_4|$ decreases by two. In contrast to Case 1, $\mathcal{A}(\mathcal{Q} \to \mathcal{P})$ may decrease due to the increase of negative loss terms.

As the conditions under which each case emerges differ across specific configurations, this phenomenon is not analyzed in general settings. We leave a precise characterization of the conditions under which increasing the pruning ratio may invalidate the progressive intensity hypothesis as an important direction for future work.

### D.7 ADDITIONAL REMARKS

**Limitations of Current Work.** We introduce a broadly applicable hypothesis that can be extended to diverse compression methods and model types across different domains. Still, we acknowledge three important limitations in our current work. First, due to the general nature of our framework, it does not provide detailed analysis for each specific combination of methods. While our hypothesis captures high-level trends, it does not define the best compression sequence for individual cases. This motivates research into discovering the best compression orderings under practical scenarios. Second, our study is limited to joint model compression in plug-and-play settings where methods are combined post-hoc. As demands for higher compression grow, integrated design strategies should be investigated beyond simple combinations. Lastly, our framework does not yet provide explicit predictive rules or precise estimation of how much better one compression order is than another. Capturing the nonlinear and cross-layer effects required for precise sign or value prediction of compression-order advantage $\mathcal{A}(\cdot)$

remains an open problem. We therefore consider the development of predictive models for $\mathcal{A}$ and meta-learning approaches for automatic order selection as a promising direction for future research.

**Future Work.** In addition to addressing the aforementioned limitations, future directions may also include extensions of our current framework. First, a systematic study of interference across different pipeline designs would provide deeper insights beyond our current empirical findings. Another direction is to automate compression order selection based on observed trends. A unified approach that generalizes across cases may offer a better understanding on the role of compression order. Lastly, evaluating our hypothesis on emerging architectures such as Mixture-of-Experts and multimodal LLMs may broaden its generality.

**Usage of AI Assistants.** We employ ChatGPT[3] (GPT-4o) and Perplexity[4] exclusively for language polishing purposes; for improving grammar and clarity at the sentence level. We do not use them for any research-related tasks, including code implementation, theoretical derivation, and result analysis.

---

[3]`https://chatgpt.com/`
[4]`https://www.perplexity.ai/`

