# OpenReview forum: "Prune-then-Quantize or Quantize-then-Prune? Understanding the Impact of Compression Order in Joint Model Compression"
_ICLR.cc/2026/Conference — ICLR 2026 Poster_

### Official Review · Reviewer_BEC6 · 2025-10-29

**Soundness:** 3
**Presentation:** 2
**Contribution:** 2
**Rating:** 4
**Confidence:** 4

**Summary:**

This paper investigates the interaction between two common model compression techniques — pruning and quantization — and specifically asks whether the order of application (Prune→Quantize vs. Quantize→Prune) affects the final model performance.
The authors formalize this question by defining the compression-order advantage
$A(Q → P)=M((P ∘Q)(φ))−M((Q ∘ P)(φ))$,
where M(⋅) denotes model performance. They also introduce the Compression-Equivalent Ratio $C_P^*$ , the quantization ratio that produces equivalent compression to a given pruning ratio so that both methods can be easily compared/combined under a unified metric.
Through a series of controlled experiments across several architectures and datasets, the paper finds that the order of operations does matter if one wishes to preserve the maximum model accuracy

**Strengths:**

1.	Clear and focused research question.
The “prune–then–quantize or quantize–then–prune?” question is simple but practically relevant for model deployment pipelines.
2.	Well-defined theoretical framework.
The introduction of the compression-order advantage $A(Q → P)$ and compression-equivalent ratio $C_p^*$  gives a clear formal basis for comparing compression orders.
3.	Systematic experimentation and consistency in results.
The experiments are comprehensive, covering multiple pruning ratios and quantization levels, with consistent definitions across models. And $A(Q→ P)$ behavior with $\Delta C$

**Weaknesses:**

1.	The compression order advantage $A$ was empirically measured for a variety of scenarios and it’s trends with $C_p^* - C_Q$ was demonstrated. But the actual value of $A$, and more specifically whether it is positive or negative, cannot be determined by the mere $C_p^*$ and $C_Q$ values. So I don’t understand how can a user decide which method to apply first,
2.	I don’t fully understand the practical meaning of "well-designed pruning" in Assumption 2. “The pruning method is chosen to minimize performance degradation”.
Isn’t that conceptually the goal of any pruning methods?
From your phrasing it seems that there is only "one" that minimize the performance degradation. Yet, THERE ARE many pruning methods (also in your experiments). So which one of these DID minimize the performance degradation?

**Questions:**

$Q_1$. Figures 3, 4 and 6. It is unclear at first glance why multiple points appear for a single pruning ratio or why some curves are nearly flat around zero. I figure that the different points in each curve correspond to different compression bitwidth. Please clarify it in text for a more fluent reading (e.g., the different points in each curve correspond to $C_Q=$a,b,c,d, and e) IMHO it will make it more fluent for readers

$Q_2$. $C_p^* $ is defined for a vanilla model. That is, apply the pruning, measure the model’s accuracy and find the compression ratio $C_p^* $ of the quantization that leads to the same accuracy. I understand the reasoning for this methodology. You want to have a common metric so that you can have a proper $x$ axis.
But what happens when you reverse the order of compressions?
Quantization, if it is applied as the second compression, still compresses by $C_Q$ (Does it?). However, I’m not sure if pruning still achieves the same analogous compression rate $C_p^*$…

$Q_3$. How does this methodology can be used for other compression methods such as joint-compression (e.g., merging similar tokens in KV-Cache, merging heads in Attention, etc.) or Low rank approximations (LoRA).

$Q_4$. Can you give reference to the assumptions in Assumption 1.

$Q_5$. Figure 5 and related text. Can you clarify? The rotation without quantization did not introduce any accuracy degradation. Am I right? Only the pruning on top of the rotation introduced an increased degradation. Is it because you prune additional weights that did not exist prior to rotation?

$Q_6$. Table 1. I guess the $C_Q$ stems from quantization of 16bit to 4,5,…9bit. Please add it to make the table clearer

$Q_7$. Make the result simpler to read. For example, in Finding 5 which you put (wisely) within a frame so that any reader can take-away this message. I think you mean “Quatization and then pruning achieves higher accuracy (than pruning and then quantization)”. Add such simpler phrasing, so that anyone, even one that did not read your hypothesis can take-away your finding.

---

> ### Author Response · Authors · 2025-11-21
> **Rebuttal by Authors (1)**
>
> We deeply appreciate the reviewer’s careful assessment and thoughtful review.
> We have thoroughly incorporated your suggestions and provided detailed responses to every point.
> Please kindly refer to the revised manuscript, where updated contents are highlighted in blue for clarity.
>
> ### **[W1] User decision based on compression order advantage.**
>
> **[A1]**
> As the reviewer pointed out, we agree that the CER difference alone cannot fully predict the exact value or sign of the compression order advantage $\mathcal{A}(\cdot)$.
> Rather than serving as an exact prediction rule for each individual configuration, we emphasize that CER functions as a high-level guiding principle that consistently captures general trends across diverse settings.
>
> The main contribution of our work is not to predict $\mathcal{A}$ for each specific combination, but to analyze the structural effect of compression order and demonstrate, both theoretically and empirically, that applying weaker compression first is generally advantageous.
> However, as the analysis is derived under assumptions such as layer-wise independence and well-designed compression, non-linear and cross-layer effects in practical networks inevitably introduce noise that prevents precise sign prediction of $\mathcal{A}$.
> Nevertheless, our experiments consistently reveal similar qualitative trends across diverse models, compression methods, and multi-stage pipelines.
> In addition, as Reviewer UfHH pointed out, our work also reports several useful secondary findings—for example, that rotation exacerbates pruning effects (Figure 5), that structured and unstructured pruning differ in their interference behavior (Table 1), and that vision models exhibit stronger order sensitivity than LLMs (Finding 5).
> We believe that this combination of primary and secondary evidence may offer actionable guidelines for practitioners when deciding which compression order to adopt.
>
> At the same time, we agree with the reviewer that offering explicit predictive rules or accurately forecasting how much better one order is than another remains the final goal of this line of research.
> Developing predictive models for $\mathcal{A}$ or meta-learning-based order selection represents a promising direction toward achieving this long-term objective.
> Motivated by this feedback, we expanded the discussion in Appendix D.7 (Limitations of Current Work) to articulate these challenges and future directions.
>
> ### **[W2] Practical meaning of “well-designed pruning”.**
>
> **[A2]**
> We clarify that Assumption 2 is not intended to single out a unique pruning algorithm but rather to formalize a minimal requirement shared across standard pruning approaches.
> The requirement specifies that pruning should be guided by a performance-preserving objective rather than operate blindly, which reflects how practical pruning methods are generally constructed.
> Our intention for introducing this assumption is to exclude objective-free algorithms such as random pruning, whose behavior is incompatible with any practical algorithms.
>
> As the reviewer implied, minimizing performance degradation is indeed the shared goal of most pruning methods.
> Hence, “well-designed pruning” serves a conventional and broadly accepted assumption that abstracts this common goal without committing to any specific method.
> Our experiments include multiple pruning methods such as SparseGPT [1] and Wanda [2], all of which align with the assumption’s requirement.
> In conclusion, the assumption remains agnostic to which method performs the best; it merely filters out unreasonable strategies that would fall outside the scope of tractable analysis.
> Motivated by the reviewer’s insight, we updated Assumption 2 to eliminate any potential implication of invoking a ‘single’ optimal pruning method.
>
> ### **[Q1/Q6] Clarification of experiment settings (Table 1, Figures 3, 4, and 6).**
>
> **[A3]**
> We appreciate the careful reading and agree that providing additional clarification by text would help readers to grasp the underlying intuition more easily.
> In Figures 3, 4, and 6, each point corresponds to a specific compression-ratio pair, defined by pruning ratio $p \in [0.05, 0.1, 0.15, 0.2, 0.25, 0.3]$ and quantization bit-width $B_{\mathcal{Q}} \in [4, 5, 6, 7, 8]$.
> For Table 1, we further clarify that the quantization component considers bit-widths $B_{\mathcal{Q}} \in [4, 5, 6, 7, 8, 9]$.
> As LLMs are known to be highly robust to quantization, some curves—especially those at lower joint compression intensities—may appear nearly flat, since the performance degradation (including bitwidth-induced variation) becomes extremely small.
> Following the reviewer’s suggestions, we have updated Table 1 and Section 5.2 to explicitly state these experiment settings and improve the readability of both the figures and the table.

---

> ### Author Response · Authors · 2025-11-21
> **Rebuttal by Authors (2)**
>
> ### **[Q2] Order-invariance of compression-equivalent ratio.**
>
> **[A4]**
> As the reviewer pointed out, we agree that the definition and interpretation of Compression Equivalent Ratio (CER) $C_{\mathcal{P}}^{* }$ should be explicitly distinguished from how pruning behaves when it appears as the second step in a joint pipeline.
>
> - **Definition of $C_{\mathcal{P}}^{*}$.**
> As defined in the paper, $C_{\mathcal{P}}^{* }$ is always computed on the original (uncompressed or vanilla) model $\phi$ and reflects the intrinsic intensity of a pruning configuration (when it is applied independently).
> Specifically, for a given pruning ratio $C_{\mathcal{P}}$, we evaluate pruning alone on the original model and identify the quantization ratio $C_{\mathcal{Q}}^{’}$ that satisfies $\mathcal{M}(\mathcal{Q}(\phi; C_{\mathcal{Q}}^{’}))=\mathcal{M}(\mathcal{P}(\phi; C_{\mathcal{P}}))$.
> Following Definition 3, we set $C_{\mathcal{P}}^{* }=C_{\mathcal{Q}}^{’}$.
> This procedure establishes a shared scale of compression ratio between pruning and quantization, which does not involve any intermediate compressed models.
>
> - **Order and compression ratios.**
> Both pruning and quantization have order-invariant compression ratios $C_{\mathcal{P}}$ and $C_{\mathcal{Q}}$ (refer to Section 2 for details).
> Pruning removes the same proportion $p$ of parameters regardless of whether it is applied  first or second; thus its compression ratio $C_{\mathcal{P}} = 1 / (1-p)$ remains unchanged.
> Similarly, the quantization bit-width determines a fixed $C_{\mathcal{Q}}={B_{orig}} / B_{\mathcal{Q}}$ independently of the compression order ($B_{orig}$ and $B_{\mathcal{Q}}$ are the original and target bit-widths, respectively).
>
> - **Interpretation of CER under order switching.**
> In Figures 3 to 7, the horizontal axis $C_{\mathcal{P}}^{* } - C_{\mathcal{Q}}$ should therefore be viewed as a comparison of intrinsic strengths between pruning $\mathcal{P}(\cdot)$ and quantization $\mathcal{Q}(\cdot)$, calibrated on the original model.
> The vertical axis then measures the difference in performance between the two orders at the same intrinsic intensities.
> Importantly, we do not assume that pruning and quantization, when placed second in the joint pipeline, behaves identically to applying it on the original model.
> Instead, $C_{\mathcal{P}}^{*}$ and $C_{\mathcal{Q}}$ are kept fixed as intrinsic reference scales, and the order advantage reflects the interaction between the two methods.
>
> Following the reviewer’s suggestion, we have revised Section 5.2 to clearly distinguish intrinsic intensity (defined on the original model) from order-dependent interaction (evaluated in joint compression).
>
> ### **[Q3] Generalization towards other compression techniques.**
>
> **[A5]**
> We agree that examining whether the Progressive Intensity Hypothesis extends to other compression techniques such as Parameter Efficient Fine-Tuning (PEFT; e.g., LoRA) and parameter sharing is crucial for demonstrating its generalizability.
> Following the reviewer’s suggestion, we conducted additional experiments on two compression combinations: 1) Pruning + (Quantization with PEFT), and 2) Parameter Sharing + Pruning.
> We updated Section 5.4 with these new results (Figures 8 and 9).
>
> - **PEFT results.**
> We evaluate the combination of SparseGPT (pruning) with RTN (quantization) + LoRA [18] (PEFT) on LLaMA 3 models (Figure 8).
> Knowledge distillation fundamentally differs from traditional compression methods as it involves training that may recover performance, necessitating its integration with other compression techniques to function as a holistic model compression approach that reduces model size or accelerates inference while minimizing performance degradation.
> Our experiments combining quantization with LoRA fine-tuning reveal that while subsequent fine-tuning mitigates compression errors to some extent, the Progressive Intensity Hypothesis still holds robustly.
>
> - **Parameter sharing results.**
> We also validate our hypothesis on parameter sharing, a compression technique that reduces model size by sharing weight parameters across different network components.
> Figure 9 reports experimental results combining Basis Sharing [19] with magnitude-based pruning, demonstrating that the compression order advantage $\mathcal{A}(\cdot)$ increases monotonically with the CER difference, consistent with our pruning-quantization findings.
>
> These two sets of experiments demonstrate that the Progressive Intensity Hypothesis extends beyond pruning-quantization combinations to a broader spectrum of model compression methods, capturing a fundamental principle governing sequential compression pipelines regardless of the specific compression mechanisms employed.
> This generalizability underscores the practical utility of our framework for designing efficient multi-method compression strategies across diverse compression paradigms.

---

> ### Author Response · Authors · 2025-11-21
> **Rebuttal by Authors (3)**
>
> ### **[Q4] Justification of Assumption 1.**
>
> **[A6]**
> Assumption 1 draws upon well-established principles in neural network compression, with both its layer-wise independence and error-performance trade-off components consistently employed throughout the literature.
> Notably, similar assumptions have been standard practice in model compression research, appearing in both foundational works and recent large-scale studies.
> The following examples show that widely used pruning and quantization algorithms also rely on Assumption 1.
>
> - **Optimal Brain Surgeon (OBS) [3] and its variants.**
> The OBS framework formulates independent layer contributions and exploits second-order derivative information to evaluate weight importance for pruning.
> As variants of OBS, recent pruning (SparseGPT [1], OBA [4]) and quantization (OBQ [5], OPTQ [6]) algorithms also inherit the same layer-wise assumptions.
>
> - **Foundational quantization methods for CNNs.**
> Classical CNN quantization methods also rely heavily on layer-wise formulation.
> For instance, AdaRound [7] performs per-layer rounding optimization, while BRECQ [8] reconstructs blocks through local second-order approximations.
> QDrop [9] similarly adopts a layer-level robustness approach by stochastically dropping quantization during local reconstruction.
>
> - **Modern quantization methods for ViTs.**
> Despite the complexity of ViTs, recent approaches frequently rely on layer-wise formulations and, in some cases, further simplify Attention modules into linear layers to enable feasible optimization.
> For instance, APHQ-ViT [10] adopts layer-wise reconstruction using Hessian-based perturbations, and FIMA-Q [11] leverages Fisher information approximations under a similar abstraction.
> LampQ [12] even extends this principle further by performing mixed-precision quantization relying on the local error of each layer.
>
> - **State-of-the-art quantization methods for LLMs.**
> Leading quantization approaches for LLMs similarly adopt layer-wise abstraction for efficient compression.
> OmniQuant [13] targets layer-wise reconstruction error as its core objective and reduces quantization error through block-level optimization.
> EfficientQAT [14] achieves the state-of-the-art result in quantization-aware training by leveraging layer-wise independence while employing reconstruction loss as an efficient proxy for preserving model performance.
>
> - **Current pruning methods for LLMs.**
> Modern pruning techniques for large language models demonstrate the continued relevance of layer-wise independence.
> LLM-Pruner [15] performs structured compression by analyzing per-layer gradients to detect removable coupled components.
> Wanda [2] exemplifies efficient layer-wise pruning by using weight-activation products as sensitivity metrics per output.
> K-prune [16] achieves retraining-free compression through layer-wise parameter importance estimation.
>
> - **Existing joint compression studies.**
> Existing works (Harma et al. [17]) are similarly grounded in layer-wise modeling for theoretical analysis.
>
> Taken together, these examples show that our assumptions are broadly adopted in the model compression community for deriving theoretical insights and practical algorithms.
>
> Empirically, our experimental results confirm that the simplified model reflects actual sensitivity trends observed in practice.
> Furthermore, EfficientQAT provides direct evidence supporting Assumption 1 by demonstrating that layer-wise approximation errors remain manageable in practice.
> For instance, on LLaMA 2 7B (WikiText-2 baseline perplexity: 5.47), naive 2-bit RTN quantization causes catastrophic collapse (WikiText-2 perplexity > 1,000), yet block-wise reconstruction training alone recovers performance to 7.79 and subsequent end-to-end refinement achieves 6.86.
> This dramatic recovery suggests that discrepancy introduced by layer-wise independence is relatively minor in comparison; the practical impact of violating full inter-layer modeling is limited.
>
> Overall, these assumptions are consistent with both prior literature and empirical observations.

---

> ### Author Response · Authors · 2025-11-21
> **Rebuttal by Authors (4)**
>
> ### **[Q5] Clarification of Finding 3 (Rotation-Pruning Interaction).**
>
> **[A7]**
> We appreciate the reviewer’s attention to our observation regarding the effect of rotation in pruning-only scenarios.
> Our results in Figure 5 indicate that the rotated model (without quantization) exhibits a noticeably larger performance drop under pruning compared to the non-rotated baseline.
> As further discussed in Appendix D.4 and shown in Figure 15, pruning a rotated model may introduce two types of error: matrix-wise error (due to residual rotation components) and instance-wise error (due to altered pruning decisions).
> We acknowledge that the reviewer’s intuition aligns with our findings: rotation alone does not degrade performance, as it is an orthogonal transformation, but pruning on top of rotation may introduce extra degradation.
> To make this clearer, we revised Section 5.2 (specifically the part corresponding to Finding 3) to explicitly articulate the two error sources introduced by rotation prior to pruning.
> We hope the updated manuscript, along with our detailed explanation, clarify the interplay between rotation and pruning.
>
> ### **[Q7] More intuitive takeaways for experimental results.**
>
> **[A8]**
> We thank the reviewer for emphasizing the importance of improving the accessibility of our key results and ensuring that the take-away messages are more intuitive for broader readers.
> We agree that the initial phrasings of Findings 1, 5, 6, and 7 rely on earlier hypotheses, and can be simplified to help readers grasp the core result without needing to look back.
> To reflect the reviewer’s advice, we modified the corresponding Findings to explicitly state their high-level takeaways (note that the former Finding 7 is now renumbered as Finding 9 after incorporating the quantization-aware training results).
> We believe the updated phrasings will enable readers to understand the key insights more easily.
>
> ### **References.**
>
> [1] Frantar et al., “SparseGPT: Massive Language Models Can Be Accurately Pruned in One-Shot”, ICML 2023
>
> [2] Sun et al., “A Simple and Effective Pruning Approach for Large Language Models”, ICLR 2024
>
> [3] Hassibi et al., “Second order derivatives for network pruning: Optimal Brain Surgeon”, NeurIPS 1992
>
> [4] Sun et al., “Optimal Brain Apoptosis”, ICLR 2025
>
> [5] Frantar et al., “Optimal Brain Compression: A Framework for Accurate Post-Training Quantization and Pruning”, NeurIPS 2022
>
> [6] Frantar et al., “GPTQ: Accurate Post-Training Quantization for Generative Pre-trained Transformers”, ICLR 2023
>
> [7] Nagel et al., “Up or Down? Adaptive Rounding for Post-Training Quantization”, ICML 2020
>
> [8] Li et al., “BRECQ: Pushing the Limit of Post-Training Quantization by Block Reconstruction”, ICLR 2021
>
> [9] Wei et al., “QDrop: Randomly Dropping Quantization for Extremely Low-bit Post-Training Quantization”, ICLR 2022
>
> [10] Wu et al., “APHQ-ViT: Post-Training Quantization with Average Perturbation Hessian Based Reconstruction for Vision Transformers”, CVPR 2025
>
> [11] Wu et al., “FIMA-Q: Post-Training Quantization for Vision Transformers by Fisher Information Matrix Approximation”, CVPR 2025
>
> [12] Kim et al., “LampQ: Towards Accurate Layer-wise Mixed Precision Quantization for Vision Transformers”, AAAI 2026
>
> [13] Shao et al., “OmniQuant: Omnidirectionally Calibrated Quantization for Large Language Models”, ICLR 2024
>
> [14] Chen et al., “EfficientQAT: Efficient Quantization-Aware Training for Large Language Models”, ACL 2025
>
> [15] Ma et al., “LLM-Pruner: On the Structural Pruning of Large Language Models”, NeurIPS 2023
>
> [16] Park et al., “Accurate Retraining-free Pruning for Pretrained Encoder-based Language Models”, ICLR 2024
>
> [17] Harma et al., “Effective Interplay between Sparsity and Quantization: From Theory to Practice”, ICLR 2025
>
> [18] Hu et al., “LoRA: Low-Rank Adaptation of Large Language Models”, ICLR 2022
>
> [19] Wang et al., “Basis Sharing: Cross-Layer Parameter Sharing for Large Language Model Compression”, ICLR 2025

---

> ### Author Response · Authors · 2025-11-28
> **A Gentle Reminder by the Authors**
>
> Dear Reviewer BEC6,
>
> We deeply value the effort and attention you have given to your review of our submission.
>
> As the discussion phase approaches its end, we would like to gently remind that only **five** days remain for additional comments or questions.
> We would be grateful for the opportunity to address any further concerns you may wish to raise before the discussion period concludes.
>
> With sincere gratitude,
>
> Authors of Submission 16763

---

### Official Review · Reviewer_1tMH · 2025-10-30

**Soundness:** 3
**Presentation:** 4
**Contribution:** 3
**Rating:** 8
**Confidence:** 2

**Summary:**

This paper investigates how the order of compression operations (e.g., pruning and quantization) affects model performance. The authors formalize the ordering problem, propose quantitative metrics, and validate the intuitive Progressive Intensity Hypothesis—that weaker perturbations should be applied before stronger ones. The paper presents both theoretical insights and extensive experiments on language and vision models, confirming the hypothesis and providing clear practical guidance for compression practitioners.

**Strengths:**

1. The topic is practical and relevant to real-world model deployment.
2. Experiments are extensive, covering multiple models and compression settings.
3. The paper provides a clear and actionable guideline that practitioners can easily adopt.
4. The writing is clear and the code is open-source.

**Weaknesses:**

The paper mainly evaluates compression pipelines that consist of two or three stages (e.g., prune–quantize or prune–quantize–prune). Based on the proposed theory, could this framework be extended to longer or more complex multi-stage compression pipelines, and would the same hypothesis still lead to performance gains compared with existing setups/shorter pipelines?

**Questions:**

Please see the weakness.

---

> ### Author Response · Authors · 2025-11-21
> **Rebuttal by Authors**
>
> We deeply appreciate the reviewer’s careful assessment and thoughtful review.
> We have thoroughly incorporated your suggestions and provided detailed responses to every point.
> Please kindly refer to the revised manuscript, where updated contents are highlighted in blue for clarity.
>
> ### **[W1] Theoretical extension towards more longer/complex multi-stage compression pipelines.**
>
> **[A1]**
> We appreciate the reviewer's insightful question regarding the extensibility of our framework.
> We would like to address this concern from two perspectives: (1) extension to longer multi-stage pipelines, and (2) generalization to compression techniques beyond pruning-quantization combinations.
>
> Regarding longer multi-stage pipelines, we provide both the theoretical foundation and empirical validation needed to support long compression sequences.
> The n-method ordering theorem in Appendix B.3 is derived through recursive application of pairwise analysis, which is the structure mirrored in our multi-stage experiments.
> Our multi-stage experiments in Section 5.4 (Figure 7) can be systematically decomposed into sequential pairwise comparisons, and their consistent patterns across compression ratios align closely with the theoretical n-method prediction.
>
> Regarding more complex setups beyond pruning-quantization, we agree that examining whether the Progressive Intensity Hypothesis extends to other compression techniques such as Parameter Efficient Fine-Tuning (PEFT; e.g., LoRA) and parameter sharing is crucial for demonstrating its generalizability.
> Following the reviewer's suggestion, we conducted additional experiments on two compression combinations: 1) Pruning + (Quantization with PEFT), and 2) Parameter Sharing + Pruning.
> We updated Section 5.4 with these new results (Figures 8 and 9).
>
> - **PEFT results.**
> We evaluate the combination of SparseGPT (pruning) with RTN (quantization) + LoRA [1] (PEFT) on LLaMA 3 models (Figure 8).
> Knowledge distillation fundamentally differs from traditional compression methods as it involves training that may recover performance, necessitating its integration with other compression techniques to function as a holistic model compression approach that reduces model size or accelerates inference while minimizing performance degradation.
> Our experiments combining quantization with LoRA fine-tuning reveal that while subsequent fine-tuning mitigates compression errors to some extent, the Progressive Intensity Hypothesis still holds robustly.
>
> - **Parameter sharing results.**
> We also validate our hypothesis on parameter sharing, a compression technique that reduces model size by sharing weight parameters across different network components.
> Figure 9 reports experimental results combining Basis Sharing [2] with magnitude-based pruning, demonstrating that the compression order advantage $\mathcal{A}(\cdot)$ increases monotonically with the CER difference, consistent with our pruning-quantization findings.
>
> These two sets of experiments demonstrate that the Progressive Intensity Hypothesis extends beyond pruning-quantization combinations to a broader spectrum of model compression methods, capturing a fundamental principle governing sequential compression pipelines regardless of the specific compression mechanisms employed.
>
> We hope these clarifications and additional experiments address the reviewer's concerns and demonstrate both the theoretical soundness and practical applicability of our framework for extended and diverse compression pipelines.
>
> ### **References.**
>
> [1] Hu et al., “LoRA: Low-Rank Adaptation of Large Language Models”, ICLR 2022
>
> [2] Wang et al., “Basis Sharing: Cross-Layer Parameter Sharing for Large Language Model Compression”, ICLR 2025

---

> ### Author Response · Authors · 2025-11-28
> **A Gentle Reminder by the Authors**
>
> Dear Reviewer 1tMH,
>
> We deeply value the effort and attention you have given to your review of our submission.
>
> As the discussion phase approaches its end, we would like to gently remind that only **five** days remain for additional comments or questions.
> We would be grateful for the opportunity to address any further concerns you may wish to raise before the discussion period concludes.
>
> With sincere gratitude,
>
> Authors of Submission 16763

---

### Official Review · Reviewer_k2Nf · 2025-10-31

**Soundness:** 3
**Presentation:** 3
**Contribution:** 2
**Rating:** 4
**Confidence:** 3

**Summary:**

This paper investigates the importance of compression order in joint model compression (e.g., pruning and quantization) and proposes the "Progressive Intensity Hypothesis" (PIH). The hypothesis posits that applying weaker perturbations (compressions) first, followed by stronger ones, leads to superior model performance .

**Strengths:**

The paper is well-organized and easy to understand.

The paper addresses a practical yet underexplored problem in model compression: the order of joint compression.

**Weaknesses:**

1. The paper introduces the "Compression Equivalent Ratio" (CER) to unify the "intensity" metric. However, calculating the CER for a pruning method requires running the pruning experiment independently to measure its performance (e.g., 65% accuracy), and then finding (or interpolating) the quantization ratio $\mathcal{Q}$ that yields the same performance. This implies that to apply the hypothesis, one must first run all compression methods individually to determine their "intensity" ranking. It is questionable whether the computational cost of this ranking process is substantially lower than the cost of directly testing both compression orders (e.g., $\mathcal{P} \rightarrow \mathcal{Q}$ and $\mathcal{Q} \rightarrow \mathcal{P}$).

2. The assumptions for the theoretical analysis are an oversimplification of neural networks. The paper assumes "Layer-wise independence" , an "Error-performance trade-off"  (both in Assumption 1), and "Disjoint Selectivity" (Definition 5). These assumptions largely ignore the correlations between layers and the different sensitivities of various layer types.

3. The scope of the hypothesis is mainly focused on "post-hoc" compression settings. Additionally, its core analysis revolves almost exclusively around pruning and quantization scenarios. It is unclear how well the hypothesis generalizes to combinations involving other compression techniques (like knowledge distillation or parameter sharing).

**Questions:**

see weakness

---

> ### Author Response · Authors · 2025-11-21
> **Rebuttal by Authors (1)**
>
> We deeply appreciate the reviewer’s careful assessment and thoughtful review.
> We have thoroughly incorporated your suggestions and provided detailed responses to every point.
> Please kindly refer to the revised manuscript, where updated contents are highlighted in blue for clarity.
>
> ### **[W1] Computational cost of ranking process.**
>
> **[A1]**
> We agree with the reviewer’s concern about the computational cost and offer further clarification on this issue.
> Our study focuses on joint compression, which integrates multiple existing techniques in a post-hoc fashion.
> This setup naturally assumes that the baseline results for each standalone method are already available in practice.
> These baselines are generally accessible for most compression methods, which can be directly incorporated into CER calculation.
> Still, we agree that in exceptional cases where no baseline results exist, the cost of CER estimation and direct order testing may possibly become comparable.
>
> However, in practical settings, directly evaluating joint compression requires substantial additional engineering effort to integrate multiple methods sequentially, far exceeding the cost of simply running publicly available implementations.
> Moreover, such integration is not reusable when exploring different combinations of compressed methods, whereas our CER-based comparison offers a clear advantage by enabling intensity assessment without any combined implementation.
> In addition, considering experiment count only, CER needs two runs (the standalone baselines), while direct comparison requires four runs (each method applied in both orders), resulting in roughly twice the cost.
>
> In summary, CER offers a unified way to compare compression intensity in joint compression scenarios, and we consider it both efficient and practical in general contexts.

---

> ### Author Response · Authors · 2025-11-21
> **Rebuttal by Authors (2)**
>
> ### **[W2] Strong theoretical assumptions.**
>
> **[A2]**
> We agree that Assumption 1 may appear oversimplified, as it does not explicitly capture inter-layer dependencies or layer-type heterogeneity in modern neural networks.
> However, our theoretical framework deliberately adopts these assumptions to achieve two critical goals:
> 1. **Theoretical tractability.**
> The non-linearity and architectural depth of neural networks induce complex inter-layer dependencies that make it infeasible to directly analyze compression effects under a full model formulation.
> The layer-wise assumption alleviates this difficulty by treating negative reconstruction error as a proxy for performance, enabling a tractable and interpretable analysis of compression behavior.
> 2. **Unified abstraction.**
> Rather than modeling every architectural detail, we aim to establish a general principle applicable across diverse compression methods, model architectures, and domains.
> This abstraction allows us to understand effects of compression order under a unified view, which would be impossible with method-specific or architecture-specific assumptions.
>
> Notably, similar assumptions have been standard practice in model compression research, appearing in both foundational works and recent large-scale studies.
> The following examples show that widely used pruning and quantization algorithms rely also on Assumption 1.
>
> - **Optimal Brain Surgeon (OBS) [1] and its variants.**
> The OBS framework formulates independent layer contributions and exploits second-order derivative information to evaluate weight importance for pruning.
> As variants of OBS, recent pruning (SparseGPT [2], OBA [3]) and quantization (OBQ [4], OPTQ [5]) algorithms also inherit the same layer-wise assumptions.
>
> - **Foundational quantization methods for CNNs.**
> Classical CNN quantization methods also rely heavily on layer-wise formulation.
> For instance, AdaRound [6] performs per-layer rounding optimization, while BRECQ [7] reconstructs blocks through local second-order approximations.
> QDrop [8] similarly adopts a layer-level robustness approach by stochastically dropping quantization during local reconstruction.
>
> - **Modern quantization methods for ViTs.**
> Despite the complexity of ViTs, recent approaches frequently rely on layer-wise formulations and, in some cases, further simplify Attention modules into linear layers to enable feasible optimization.
> For instance, APHQ-ViT [9] adopts layer-wise reconstruction using Hessian-based perturbations, and FIMA-Q [10] leverages Fisher information approximations under a similar abstraction.
> LampQ [11] even extends this principle further by performing mixed-precision quantization relying on the local error of each layer.
>
> - **State-of-the-art quantization methods for LLMs.**
> Leading quantization approaches for LLMs similarly adopt layer-wise abstraction for efficient compression.
> OmniQuant [12] targets layer-wise reconstruction error as its core objective and reduces quantization error through block-level optimization.
> EfficientQAT [13] achieves the state-of-the-art result in quantization-aware training by leveraging layer-wise independence while employing reconstruction loss as an efficient proxy for preserving model performance.
>
> - **Current pruning methods for LLMs.**
> Modern pruning techniques for large language models demonstrate the continued relevance of layer-wise independence.
> LLM-Pruner [14] performs structured compression by analyzing per-layer gradients to detect removable coupled components.
> Wanda [15] exemplifies efficient layer-wise pruning by using weight-activation products as sensitivity metrics per output.
> K-prune [16] achieves retraining-free compression through layer-wise parameter importance estimation.
>
> - **Existing joint compression studies.**
> Existing works (Harma et al. [17]) are similarly grounded in layer-wise modeling for theoretical analysis.
>
> Taken together, these examples show that our assumptions are broadly adopted in the model compression community for deriving theoretical insights and practical algorithms.
>
> Empirically, our experimental results confirm that the simplified model reflects actual sensitivity trends observed in practice.
> Furthermore, EfficientQAT provides direct evidence supporting Assumption 1 by demonstrating that layer-wise approximation errors remain manageable in practice.
> For instance, on LLaMA 2 7B (WikiText-2 baseline perplexity: 5.47), naive 2-bit RTN quantization causes catastrophic collapse (WikiText-2 perplexity > 1,000), yet block-wise reconstruction training alone recovers performance to 7.79 and subsequent end-to-end refinement achieves 6.86.
> This dramatic recovery suggests that discrepancy introduced by layer-wise independence is relatively minor in comparison; the practical impact of violating full inter-layer modeling is limited.
>
> Overall, these assumptions enable principled theoretical analysis while remaining consistent with both prior literature and empirical observations.

---

> ### Author Response · Authors · 2025-11-21
> **Rebuttal by Authors (3)**
>
> ### **[W3] Generalization towards other compression techniques.**
>
> **[A3]**
> We agree that examining whether the Progressive Intensity Hypothesis extends to other compression techniques such as Parameter Efficient Fine-Tuning (PEFT; e.g., LoRA) and parameter sharing is crucial for demonstrating its generalizability.
> Following the reviewer’s suggestion, we conducted additional experiments on two compression combinations: 1) Pruning + (Quantization with PEFT), and 2) Parameter Sharing + Pruning.
> We updated Section 5.4 with these new results (Figures 8 and 9).
>
> - **PEFT results.**
> We evaluate the combination of SparseGPT (pruning) with RTN (quantization) + LoRA [18] (PEFT) on LLaMA 3 models (Figure 8).
> Knowledge distillation fundamentally differs from traditional compression methods as it involves training that may recover performance, necessitating its integration with other compression techniques to function as a holistic model compression approach that reduces model size or accelerates inference while minimizing performance degradation.
> Our experiments combining quantization with LoRA fine-tuning reveal that while subsequent fine-tuning mitigates compression errors to some extent, the Progressive Intensity Hypothesis still holds robustly.
>
> - **Parameter sharing results.**
> We also validate our hypothesis on parameter sharing, a compression technique that reduces model size by sharing weight parameters across different network components.
> Figure 9 reports experimental results combining Basis Sharing [19] with magnitude-based pruning, demonstrating that the compression order advantage $\mathcal{A}(\cdot)$ increases monotonically with the CER difference, consistent with our pruning-quantization findings.
>
> These two sets of experiments demonstrate that the Progressive Intensity Hypothesis extends beyond pruning-quantization combinations to a broader spectrum of model compression methods, capturing a fundamental principle governing sequential compression pipelines regardless of the specific compression mechanisms employed.
> This generalizability underscores the practical utility of our framework for designing efficient multi-method compression strategies across diverse compression paradigms.
>
> ### **References.**
>
> [1] Hassibi et al., “Second order derivatives for network pruning: Optimal Brain Surgeon”, NeurIPS 1992
>
> [2] Frantar et al., “SparseGPT: Massive Language Models Can Be Accurately Pruned in One-Shot”, ICML 2023
>
> [3] Sun et al., “Optimal Brain Apoptosis”, ICLR 2025
>
> [4] Frantar et al., “Optimal Brain Compression: A Framework for Accurate Post-Training Quantization and Pruning”, NeurIPS 2022
>
> [5] Frantar et al., “GPTQ: Accurate Post-Training Quantization for Generative Pre-trained Transformers”, ICLR 2023
>
> [6] Nagel et al., “Up or Down? Adaptive Rounding for Post-Training Quantization”, ICML 2020
>
> [7] Li et al., “BRECQ: Pushing the Limit of Post-Training Quantization by Block Reconstruction”, ICLR 2021
>
> [8] Wei et al., “QDrop: Randomly Dropping Quantization for Extremely Low-bit Post-Training Quantization”, ICLR 2022
>
> [9] Wu et al., “APHQ-ViT: Post-Training Quantization with Average Perturbation Hessian Based Reconstruction for Vision Transformers”, CVPR 2025
>
> [10] Wu et al., “FIMA-Q: Post-Training Quantization for Vision Transformers by Fisher Information Matrix Approximation”, CVPR 2025
>
> [11] Kim et al., “LampQ: Towards Accurate Layer-wise Mixed Precision Quantization for Vision Transformers”, AAAI 2026
>
> [12] Shao et al., “OmniQuant: Omnidirectionally Calibrated Quantization for Large Language Models”, ICLR 2024
>
> [13] Chen et al., “EfficientQAT: Efficient Quantization-Aware Training for Large Language Models”, ACL 2025
>
> [14] Ma et al., “LLM-Pruner: On the Structural Pruning of Large Language Models”, NeurIPS 2023
>
> [15] Sun et al., “A Simple and Effective Pruning Approach for Large Language Models”, ICLR 2024
>
> [16] Park et al., “Accurate Retraining-free Pruning for Pretrained Encoder-based Language Models”, ICLR 2024
>
> [17] Harma et al., “Effective Interplay between Sparsity and Quantization: From Theory to Practice”, ICLR 2025
>
> [18] Hu et al., “LoRA: Low-Rank Adaptation of Large Language Models”, ICLR 2022
>
> [19] Wang et al., “Basis Sharing: Cross-Layer Parameter Sharing for Large Language Model Compression”, ICLR 2025

---

> ### Author Response · Authors · 2025-11-28
> **A Gentle Reminder by the Authors**
>
> Dear Reviewer k2Nf,
>
> We deeply value the effort and attention you have given to your review of our submission.
>
> As the discussion phase approaches its end, we would like to gently remind that only **five** days remain for additional comments or questions.
> We would be grateful for the opportunity to address any further concerns you may wish to raise before the discussion period concludes.
>
> With sincere gratitude,
>
> Authors of Submission 16763

---

### Official Review · Reviewer_UfHH · 2025-11-01

**Soundness:** 4
**Presentation:** 3
**Contribution:** 3
**Rating:** 6
**Confidence:** 3

**Summary:**

This paper revisits an underexplored question in model compression: whether the order of applying pruning and quantization matter, and why? The authors formulate the joint compression order optimization problem and propose the Progressive Intensity Hypothesis (PIH), that weaker perturbations should precede stronger ones. This hypothesis  is proved under the assumptions of disjoint selectivity and well-designed compression (Theorems 1 & 2), and extend the analysis to more realistic interference cases. Empirical validation spans language models (LLaMA-2/3 7B-13B-8B) and vision models (ResNet-18, DeiT-Base), showing consistent monotonic trends where quantization-after-pruning performs worse than pruning-after-quantization. The hypothesis is further generalized to multi-stage compression and mixed-precision quantization.

**Strengths:**

1. A new but meaningful problem explored in this paper: the paper introduces a rigorous formalization of compression order optimization, a previously neglected but practically crucial dimension in joint model compression. The Progressive Intensity Hypothesis provides a simple, actionable rule with theoretical grounding and broad implications.

2. Clear theoretical analysis.  Theoretical results are clearly derived. Theorem 1 establishes performance ordering under disjoint selectivity, relating compression order advantage A to per-unit reconstruction errors. Theorem 2 proves monotonicity with respect to compression equivalent ratio (CER) differences. These analyses connect signal perturbation and model compression behavior.

3. The empirical validation covers language, vision, and general pipelines, including experiments on LLaMA and ResNet/ViT variants. The paper goes beyond two-method cases to multi-stage and mixed-precision scenarios, reinforcing the hypothesis’ generality.

4. There are also some useful secondary findings in this paper. For example, rotation exacerbates pruning effects (Fig. 5). Structured vs. unstructured pruning differ in interference behavior (Table 1). Vision models exhibit stronger order sensitivity than LLMs.

**Weaknesses:**

1. Theoretical assumptions are strong. The “well-designed” compression assumption is idealized and not always met in real pruning heuristics. The analysis could better discuss how violations (e.g., correlated layer errors, adaptive pruning schedules) affect Theorem 2. The independence assumption between layers (Assumption 1) is strong, worth empirically validating with correlation metrics.

2. While extensions to multi-stage and MPQ are demonstrated, the theoretical discussion remains pairwise. A more general n-method ordering theorem (Appendix B.3) is mentioned but not empirically validated.

3. Connection to prior composability studies. The paper could better position itself relative to recent works like SmoothQuant (Xiao et al., 2023) or Harma et al. (2025) by emphasizing differences in analytical generality rather than empirical scope alone.

**Questions:**

1. How sensitive are the results to the performance metric M(·)? Would monotonicity still hold under different metrics (e.g., loss, perplexity, BLEU)?

2. I am interested whether the Progressive Intensity Hypothesis extend to compression combinations involving low-rank (e.g., Lora) or distillation techniques?

---

> ### Author Response · Authors · 2025-11-21
> **Rebuttal by Authors (1)**
>
> We deeply appreciate the reviewer’s careful assessment and thoughtful review.
> We have thoroughly incorporated your suggestions and provided detailed responses to every point.
> Please kindly refer to the revised manuscript, where updated contents are highlighted in blue for clarity.
>
> ### **[W1] Strong theoretical assumptions.**
>
> **[A1]**
> We appreciate the reviewer’s insightful feedback regarding the strength of our theoretical assumptions.
> Below, we clarify the motivation and validity of both Assumptions 1 (layer-wise independence) and 2 (“well-designed” compression).
>
> **1. Clarifying the role of Assumption 2.**
> Assumption 2 is not intended to imply the existence of a single optimal pruning algorithm. Instead, it formalizes a minimal requirement shared across standard pruning approaches; pruning should be guided by a performance-preserving objective rather than performed blindly.
> This assumption primarily serves to exclude objective-free algorithms such as random pruning, whose behavior is incompatible with any practical algorithms.
>
> In this point of view, minimizing performance degradation is the shared goal of most pruning methods.
> Thus, “well-designed pruning” is a conventional and widely accepted abstraction that captures this shared principle without committing to any specific algorithm.
> Our experiments incorporate multiple pruning methods such as SparseGPT [1] and Wanda [2], all of which satisfy this assumption.
> Motivated by the reviewer’s comment, we updated Assumption 2 in the manuscript to prevent any unintended implication of invoking a single ‘best’ pruning method.
>
> **2. Why Assumption 1 adopts layer-wise independence.**
> We acknowledge that Assumption 1 may appear strong, as it does not explicitly model inter-layer correlations or layer-type heterogeneity present in neural networks.
> However, this simplification is deliberate and enables two essential goals in our theoretical development:
>
> 1. **Theoretical tractability.**
> The non-linearity and architectural depth of neural networks induce complex inter-layer dependencies that make it infeasible to directly analyze compression effects under a full model formulation.
> The layer-wise assumption alleviates this difficulty by treating negative reconstruction error as a proxy for performance, enabling a tractable and interpretable analysis of compression behavior.
>
> 2. **Unified abstraction.**
> Rather than modeling every architectural detail, we aim to establish a general principle applicable across diverse compression methods, model architectures, and domains.
> This abstraction allows us to understand compression order effects under a unified view, which would be impossible with method-specific or architecture-specific assumptions.
>
> **3. Assumption 1 is a common approach in model compression.**
> Layer-wise independence is not unique to our analysis; it has been fundamental to theoretical and algorithmic developments in model compression.
> Widely adopted examples include:
>
> - **OBS [1] and second-order pruning methods** (SparseGPT [2], OBA [3], OBQ [4], OPTQ [5]), which explicitly model layer-wise contributions using Hessian-based approximations.
>
> - **CNN quantization frameworks** (AdaRound [6], BRECQ [7], QDrop [8]), which optimize reconstruction at the layer or block level.
>
> - **ViT quantization approaches** (APHQ-ViT [9], FIMA-Q [10], LampQ [11]), which rely on local Hessian or Fisher approximations and even simplify attention modules into layer-wise linear forms.
>
> - **State-of-the-art LLM quantization methods** (OmniQuant [12], EfficientQAT [13]), which explicitly minimize layer-level reconstruction error.
>
> - **Current LLM pruning approaches** (LLM-Pruner [14], Wanda [15], K-prune [16]), which evaluate per-layer sensitivity or importance.
>
> - **Existing joint compression studies** (Harma et al. [17]) likewise adopt layer-wise approximations to analyze how pruning and quantization interact.
>
> Across these diverse methods, layer-wise independence serves as a practical and theoretically grounded abstraction for both analysis and algorithm design.
>
> **4. Empirical validation and the effect of assumption violations.**
> We agree with the reviewer that understanding the effect of correlated errors is important.
> Empirically, quantization results from EfficientQAT [13] indicate that layer-wise approximation errors remain manageable in practice.
> For instance, on LLaMA 2 7B (WikiText-2 baseline perplexity: 5.47), naive 2-bit RTN quantization causes catastrophic collapse (WikiText-2 perplexity > 1,000), yet block-wise reconstruction training alone recovers performance to 7.79 and subsequent end-to-end refinement achieves 6.86.
> This dramatic recovery suggests that discrepancy introduced by layer-wise independence is relatively minor in comparison; the practical impact of violating full inter-layer modeling is limited.

---

> ### Author Response · Authors · 2025-11-21
> **Rebuttal by Authors (2)**
>
> ### **[W2] Theory and empirical validations towards multi-stage applications.**
>
> **[A2]**
> We would like to clarify that the theoretical framework and the experimental results involving multiple compression methods are closely aligned.
> The n-method ordering theorem in Appendix B.3 is derived through recursive application of pairwise analysis, which is the structure mirrored in our multi-stage experiments.
> Our multi-stage and MPQ experiments in Section 5.4  can be systematically decomposed into sequential pairwise comparisons, and their consistent patterns across compression ratios align closely with the theoretical n-method prediction.
> That said, we acknowledge that this linkage may not have been immediately evident, and thus updated Section 5.4 to clarify its alignment with the formal theory in Appendix B.3.
> We hope this resolves the concern by showing that our theoretical and empirical components are tightly connected.
>
> ### **[W3] Connection to prior composability studies.**
>
> **[A3]**
> We appreciate the reviewer’s suggestion to better contextualize our contribution within prior composability studies.
> Our work addresses a fundamentally different question than SmoothQuant [18] and provides broader analytical generality than Harma et al. [17]:
>
> 1. **SmoothQuant.**
> Unlike joint compression approaches, SmoothQuant optimizes a ‘single’ compression technique (quantization) by addressing activation outliers through per-channel scaling.
> While the paper states that “SmoothQuant is orthogonal to quantization schemes”, this means that it serves as a pre-processing step that can be applied before various quantization methods.
> However, it does not examine the order-dependent interaction when combining with other compression methods such as pruning.
> In contrast, we focus on understanding how compression order affects model performance in joint compression scenarios.
>
> 2. **Harma et al.**:
> The paper is one of our direct baselines that addresses the role of compression
> order in joint compression.
> However, as discussed in Section 1 and Appendix D.5, their theoretical analysis has three critical limitations: 1) it is confined to less practical settings of magnitude-based pruning and max-scaled quantization, 2) it does not account for modern techniques such as rotation-based methods or advanced weight-update strategies,
> and 3) it concludes that pruning followed by quantization is universally optimal,
> which does not hold in practical settings with diverse compression methods.
> In contrast, we provide a general theoretical framework based on compression intensity that applies across diverse pruning and quantization methods, and validate our Progressive Intensity Hypothesis across multiple modalities, architectures, and compression methods.
>
> Following the reviewer’s feedback, we have expanded Appendix D.5 and added explicit subsections comparing our analysis with SmoothQuant and Harma et al.
> These additions clarify that our contribution lies not only in empirical breadth but also in establishing a general analytical foundation for understanding composability across diverse compression operators, whereas prior works analyze composability within algorithm-specific designs.
>
> ### **[Q1] Sensitivity of results towards performance metric M.**
>
> **[A4]**
> We agree that clarifying the dependence on the performance metric $\mathcal{M}(\cdot)$ is important for fully understanding our results.
> Our theoretical framework is metric-agnostic and requires only that $\mathcal{M}(\cdot)$ preserves a consistent monotone ordering over model quality;
> any practical metric fits directly into the analysis.
> In the main experiments for language models, we adopt negative perplexity as $\mathcal{M}(\cdot)$ because it is a standard likelihood-based continuous metric that supports stable and efficient evaluation (i.e., perplexity is defined as the exponential of the negative log-likelihood).
> However, perplexity does not cover all aspects of performance and may diverge from downstream accuracy in some cases, thus we complement it with further evaluations under alternative metrics.
>
> Across Appendices D.1 and D.3, we confirm the similar monotonicity pattern of language models under other metrics, including zero-shot accuracy on commonsense reasoning tasks and Spearman correlation.
> For vision models, we validate the trend using top-1 image-classification accuracy.
> The consistent agreement across likelihood-based, accuracy-based, and correlation-based metrics strongly suggests that monotonicity predicted by our theory is not tied to a specific choice of $\mathcal{M}(\cdot)$ and would hold under other reasonable metrics as well.

---

> ### Author Response · Authors · 2025-11-21
> **Rebuttal by Authors (3)**
>
> ### **[Q2] Generalization towards other compression techniques.**
>
> **[A5]**
> We agree that examining whether the Progressive Intensity Hypothesis extends to other compression techniques such as Parameter Efficient Fine-Tuning (PEFT; e.g., LoRA) and parameter sharing is crucial for demonstrating its generalizability.
> Following the reviewer’s suggestion, we conducted additional experiments on two compression combinations: 1) Pruning + (Quantization with PEFT), and 2) Parameter Sharing + Pruning.
> We updated Section 5.4 with these new results (Figures 8 and 9).
>
> - **PEFT results.**
> We evaluate the combination of SparseGPT (pruning) with RTN (quantization) + LoRA [18] (PEFT) on LLaMA 3 models (Figure 8).
> Knowledge distillation fundamentally differs from traditional compression methods as it involves training that may recover performance, necessitating its integration with other compression techniques to function as a holistic model compression approach that reduces model size or accelerates inference while minimizing performance degradation.
> Our experiments combining quantization with LoRA fine-tuning reveal that while subsequent fine-tuning mitigates compression errors to some extent, the Progressive Intensity Hypothesis still holds robustly.
>
> - **Parameter sharing results.**
> We also validate our hypothesis on parameter sharing, a compression technique that reduces model size by sharing weight parameters across different network components.
> Figure 9 reports experimental results combining Basis Sharing [19] with magnitude-based pruning, demonstrating that the compression order advantage $\mathcal{A}(\cdot)$ increases monotonically with the CER difference, consistent with our pruning-quantization findings.
>
> These two sets of experiments demonstrate that the Progressive Intensity Hypothesis extends beyond pruning-quantization combinations to a broader spectrum of model compression methods, capturing a fundamental principle governing sequential compression pipelines regardless of the specific compression mechanisms employed.
> This generalizability underscores the practical utility of our framework for designing efficient multi-method compression strategies across diverse compression paradigms.
>
> ### **References.**
>
> [1] Hassibi et al., “Second order derivatives for network pruning: Optimal Brain Surgeon”, NeurIPS 1992
>
> [2] Frantar et al., “SparseGPT: Massive Language Models Can Be Accurately Pruned in One-Shot”, ICML 2023
>
> [3] Sun et al., “Optimal Brain Apoptosis”, ICLR 2025
>
> [4] Frantar et al., “Optimal Brain Compression: A Framework for Accurate Post-Training Quantization and Pruning”, NeurIPS 2022
>
> [5] Frantar et al., “GPTQ: Accurate Post-Training Quantization for Generative Pre-trained Transformers”, ICLR 2023
>
> [6] Nagel et al., “Up or Down? Adaptive Rounding for Post-Training Quantization”, ICML 2020
>
> [7] Li et al., “BRECQ: Pushing the Limit of Post-Training Quantization by Block Reconstruction”, ICLR 2021
>
> [8] Wei et al., “QDrop: Randomly Dropping Quantization for Extremely Low-bit Post-Training Quantization”, ICLR 2022
>
> [9] Wu et al., “APHQ-ViT: Post-Training Quantization with Average Perturbation Hessian Based Reconstruction for Vision Transformers”, CVPR 2025
>
> [10] Wu et al., “FIMA-Q: Post-Training Quantization for Vision Transformers by Fisher Information Matrix Approximation”, CVPR 2025
>
> [11] Kim et al., “LampQ: Towards Accurate Layer-wise Mixed Precision Quantization for Vision Transformers”, AAAI 2026
>
> [12] Shao et al., “OmniQuant: Omnidirectionally Calibrated Quantization for Large Language Models”, ICLR 2024
>
> [13] Chen et al., “EfficientQAT: Efficient Quantization-Aware Training for Large Language Models”, ACL 2025
>
> [14] Ma et al., “LLM-Pruner: On the Structural Pruning of Large Language Models”, NeurIPS 2023
>
> [15] Sun et al., “A Simple and Effective Pruning Approach for Large Language Models”, ICLR 2024
>
> [16] Park et al., “Accurate Retraining-free Pruning for Pretrained Encoder-based Language Models”, ICLR 2024
>
> [17] Harma et al., “Effective Interplay between Sparsity and Quantization: From Theory to Practice”, ICLR 2025
>
> [18] Hu et al., “LoRA: Low-Rank Adaptation of Large Language Models”, ICLR 2022
>
> [19] Wang et al., “Basis Sharing: Cross-Layer Parameter Sharing for Large Language Model Compression”, ICLR 2025

---

> ### Author Response · Authors · 2025-11-28
> **A Gentle Reminder by the Authors**
>
> Dear Reviewer UfHH,
>
> We deeply value the effort and attention you have given to your review of our submission.
>
> As the discussion phase approaches its end, we would like to gently remind that only **five** days remain for additional comments or questions.
> We would be grateful for the opportunity to address any further concerns you may wish to raise before the discussion period concludes.
>
> With sincere gratitude,
>
> Authors of Submission 16763

---

### Author Response · Authors · 2025-11-23
**Update of Manuscript**

Dear reviewers,

We sincerely thank all reviewers for their insightful reviews and constructive feedback on our manuscript.

As highlighted across the reviews, our work introduces a new and practically important yet underexplored problem of joint compression order optimization (reviewers UfHH, k2Nf, 1tMH, and BEC6).
We believe that addressing this problem will yield meaningful advantages in real-world deployment scenarios, enabling more efficient, scalable, and resource-aware model compression pipelines.
The paper is presented in a clear and well-structured manner, which helps readers to follow the motivation, theoretical development, and empirical findings with ease (reviewers k2Nf and 1tMH).
We provide rigorous theoretical analysis, including the formulation of the progressive intensity hypothesis, two formal theorems, and the introduction of compression-order advantage and compression-equivalent ratio.
Taken together, these components offer a principled understanding of how and why compression order affects outcomes (reviewers UfHH and BEC6).
Our study is further strengthened by extensive and consistent empirical validations across architectures, modalities, compression configurations, and multi-stage pipelines; thereby demonstrating the broad applicability and robustness of the proposed framework (reviewers UfHH, 1tMH, and BEC6).
Additionally, our experiments reveal useful secondary insights (e.g., rotation exacerbates pruning effects) that extend beyond the primary contributions and may inspire future work on model compression (reviewer UfHH).

Following the reviewers’ suggestions, we have incorporated several improvements to our manuscript and **highlighted all modifications in blue** for easier inspection.
Below, we summarize the key changes made in the revised manuscript:

***

- **[Assumption 2] Clarification of ‘Well-designed Pruning’**
   - We revise the description to clarify that Assumption 2 formalizes a minimal requirement broadly satisfied across standard pruning algorithms, rather than prescribing a specific pruning heuristic.

- **[Section 3 and 4] Improved Conceptual Clarity and Analytical Intuition**
   - We add clearer descriptions of compression attributes and provide additional intuition for the theoretical framework to support readers’ understanding.

- **[Section 5.2] Detailed Explanation of Experiment Settings and Interpretation**
   - We expand Section 5.2 to clearly describe the experimental configurations used in Figures 3, 4, and 5  and to articulate the meaning of each plot. These additions would help readers interpret the trends, understand how compression ratios map to performance differences, and contextualize the monotonicity patterns across pruning and quantization combinations.

- **[Findings 1, 5, 6, and 9] Hypothesis-independent Presentation of Empirical Insights**
   - We revise the descriptions of Findings 1, 5, 6, and 9 to present their empirical implications in a self-contained manner. The updated text explains each finding directly in terms of observable behaviors, enabling readers to understand the significance of the results even without prior knowledge of the progressive intensity hypothesis.
- **[Section 5.4] Generalization to PEFT and Parameter Sharing**
   - We include new experiments evaluating whether the progressive intensity hypothesis extends to lightweight fine-tuning (e.g., LoRA) and parameter sharing-based compression. Results are reported in Figures 8 and 9.
- **[Appendix D.5] A Direct Comparison with Prior Methods**
   - We expand Appendix D.5 to explicitly compare our framework with prior composability studies, including SmoothQuant and Harma et al.
- **[Appendix D.7] Limitations of Current Work**
   - As noted by reviewer BEC6, precisely predicting the magnitude or sign of compression-order advantage remains an open challenge. We update the limitations section to acknowledge this explicitly and highlight it as an important direction for future work.

***

We hope that the revisions, alongside our rebuttal, fully resolve the reviewers’ concerns and enhance the manuscript’s clarity and overall robustness.
We are grateful for your time and thoughtful evaluation, and we look forward to your feedback on the updated submission.

With best gratitude,

Authors of Submission 16763

---

### Meta-Review · Area_Chair_XdhN · 2026-01-04

**Summary:**

This paper studies compression order in joint model compression. Its main contribution is the Progressive Intensity Hypothesis, suggesting weaker perturbations should precede stronger ones. The authors provide theoretical analysis and experiments across vision and language models. Reviewers raised concerns about idealized theoretical assumptions, practical utility, generality to other techniques, and presentation clarity. The authors addressed most of these concerns. While the paper still has some limitations, such as offering primarily qualitative rather than quantitative guidance, the core ideas and supporting evidence remain compelling. Taking the reviews and rebuttal together, I recommend accepting this paper.

**Reviewer Concerns:**

Reviewers raised issues on theoretical assumptions, CER utility, generality, and clarity. Most concerns were addressed with clarifications, experiments, and improved presentation. Remaining minor issues do not affect the main contributions.

**Reviewer Scores:**

Based on the rebuttal and discussion, reviewers would likely maintain or slightly increase their scores given the added experiments and clarifications.

---

### Decision · Program_Chairs · 2026-01-26

Accept (Poster)